# Myasthenia gravis-specific aberrant neuromuscular gene expression by medullary thymic epithelial cells in thymoma

Yoshiaki Yasumizu [1,2,3], Naganari Ohkura[2,4✉], Hisashi Murata[1], Makoto Kinoshita[1], Soichiro Funaki[5], Satoshi Nojima [6], Kansuke Kido[6], Masaharu Kohara[6], Daisuke Motooka[3,7], Daisuke Okuzaki [3,7], Shuji Suganami[7], Eriko Takeuchi[1], Yamami Nakamura[2], Yusuke Takeshima [2], Masaya Arai [2], Satoru Tada[1], Meinoshin Okumura[8], Eiichi Morii [6], Yasushi Shintani [5], Shimon Sakaguchi [2], Tatsusada Okuno[1✉] & Hideki Mochizuki [1,3]

Myasthenia gravis (MG) is a neurological disease caused by autoantibodies against neuromuscular-associated proteins. While MG frequently develops in thymoma patients, the etiologic factors for MG are not well understood. Here, by constructing a comprehensive atlas of thymoma using bulk and single-cell RNA-sequencing, we identify ectopic expression of neuromuscular molecules in MG-type thymoma. These molecules are found within a distinct subpopulation of medullary thymic epithelial cells (mTECs), which we name neuromuscular mTECs (nmTECs). MG-thymoma also exhibits microenvironments dedicated to autoantibody production, including ectopic germinal center formation, T follicular helper cell accumulation, and type 2 conventional dendritic cell migration. Cell–cell interaction analysis also predicts the interaction between nmTECs and T/B cells via *CXCL12-CXCR4*. The enrichment of nmTECs presenting neuromuscular molecules within MG-thymoma is further confirmed immunohistochemically and by cellular composition estimation from the MG-thymoma transcriptome. Altogether, this study suggests that nmTECs have a significant function in MG pathogenesis via ectopic expression of neuromuscular molecules.

[1] Department of Neurology, Graduate School of Medicine, Osaka University, Suita, Osaka, Japan. [2] Department of Experimental Immunology, Immunology Frontier Research Center, Osaka University, Suita, Osaka, Japan. [3] Integrated Frontier Research for Medical Science Division, Institute for Open and Transdisciplinary Research Initiatives (OTRI), Osaka University, Suita, Osaka, Japan. [4] Department of Frontier Research in Tumor Immunology, Graduate School of Medicine, Osaka University, Suita, Osaka, Japan. [5] Department of General Thoracic Surgery, Graduate School of Medicine, Osaka University, Suita, Osaka, Japan. [6] Department of Pathology, Graduate School of Medicine, Osaka University, Suita, Osaka, Japan. [7] Genome Information Research Center, Research Institute for Microbial Diseases, Osaka University, Suita, Osaka, Japan. [8] Department of General Thoracic Surgery, National Hospital Organization Osaka Toneyama Medical Center, Osaka, Japan. ✉email: nohkura@ifrec.osaka-u.ac.jp; okuno@neurol.med.osaka-u.ac.jp

Myasthenia gravis (MG) is the most common disorder of neuromuscular transmission caused by autoantibodies against the motor endplate, such as anti-acetylcholine receptor (AChR) antibodies. MG is often accompanied by thymoma, and thymoma-associated MG (TAMG) is more difficult to manage than other forms of MG because of its frequent crisis, the need for surgery, the difficulty of perioperative management and the need for intense immunotherapies[1]. As the epidemiology, 21% of MG patients experienced thymoma[2], and 25% of thymoma patients experienced MG[3], indicating that MG and abnormalities in the thymus are closely related to each other. This is also exemplified by that thymectomy is a well-established treatment for TAMG in addition to immunosuppressive treatments[4].

Abnormalities of the thymus, in which immature thymocytes differentiate into matured CD4[+] or CD8[+] T cells, are frequently associated with a variety of autoimmune diseases, such as pure red cell aplasia and Good syndrome[5]. In the thymus, T-cell maturation and selection are conducted by the interaction with antigen-presenting cells, including thymic epithelial cells (TECs), myeloid cells, and B cells[6,7]. The positive selection of functional T cells is mediated by cortical TECs (cTECs), while the negative selection of auto-reactive T cells is mediated by medullary TECs (mTECs) presenting self-antigens on MHCs. In line with the critical role of mTECs, a loss of function mutation of *AIRE*, which is an essential transcription factor for producing self-antigens in mTECs[8], causes systemic autoimmunity called Autoimmune Polyglandular Syndrome Type 1 (APS-1)[9]. In addition, dysregulation of the thymus, especially thymoma, is frequently associated not only with MG but also neurological disorders, including encephalitis, which is caused by a wide range of autoantibodies[10–15] Thus, abnormalities of the thymus are closely associated with the generation of self-reactive auto-antibodies, which result in the development of autoimmune diseases.

Given that MG is caused by self-reactive autoantibodies, MG-specific changes within thymoma may be a clue for understanding the pathogenesis of MG. It has been reported so far that the accumulation of neurofilaments, which is expressed in neurons under the normal condition, are highly detected in MG-thymoma[16]. In addition, germinal centers (GCs) and T follicular helper (T$_{FH}$) cells, both of which play critical roles in antibody production, are also enriched in MG-thymoma[17,18]. Despite the possible contribution of these changes in MG pathology, the complete picture of MG pathogenesis from abnormalities in thymoma to auto-reactive B cell maturation is still poorly understood due to intra- and inter-individual heterogeneity of the thymus.

Here, by integratively analyzing bulk and single-cell transcriptomes of MG-thymoma to identify the complex pathogenicity of MG in thymoma, we show that a distinct subpopulation of mTECs ectopically express neuromuscular-associated molecules and could contribute to the pathogenesis of MG via presenting neuromuscular molecules to self-reactive immune cells developed in thymoma.

## Results

**Ectopic expression of neuron-related molecules in MG-thymoma.** To characterize MG-specific changes in thymoma comprehensively, we first investigated gene expression profiles from surgically dissected thymoma samples enrolled by The Cancer Genome Atlas (TCGA)[19] (Fig. 1a). Of the 116 thymoma samples with RNA-seq data, 34 were complicated with MG. In the WHO classifications, which are commonly used for the thymoma staging based on histology, there are six classifications:

Type A, AB, B1-B3, and C (i.e., A, spindle cells; AB, mixed spindle cells and lymphocytes; B1, lymphocytes > epithelial cells; B2, mixed lymphocytes and epithelial cells; B3, predominant epithelial cells; and C, carcinoma). In the present dataset, MG was associated with multiple types except for type C (Supplementary Fig. 4a), with its peak at type B3 (MG complication rate: 54.5%) and B2 (53.8%), whereas a previous report observed the peak at type B2 (71.1%)[20]. When we investigated differentially expressed genes between thymoma with and without MG (Supplementary Data 1, Supplementary Fig. 4b), 93 and 91 genes were identified as upregulated and downregulated genes in MG, respectively. The upregulated genes contained neuromuscular-related molecules; *NEFM*, *RYR3*, *GABRA5*, and immunoreceptors; *PLXNB3*, *IL13RA*. We also observed a slight increase in the acetylcholine receptor *CHRNA1*, which is the main target of autoantibodies in TAMG ($log_2$ $fold$ $change = 1.07$, $P_{adj} = 0.87$ with DESeq2[21]; $P = 0.0051$ with two-sided Mann–Whitney $U$-test, Supplementary Fig. 4c).

To dissect transcriptome changes unbiasedly, we next adopted the unsupervised gene clustering approach; Weighted Gene Co-expression Network Analysis (WGCNA)[22] on the large-scale thymoma samples (Supplementary Fig. 4d). We initially constructed gene "modules", each of which were composed of a set of genes showing correlated gene expression. In the thymoma samples, seven modules consisting of 30 -1102 genes were obtained and represented by colors (Supplementary Data 2). We next investigated the association of clinical information with a representative gene expression of each module, calculated as eigengene (Fig. 1b). MG had a most significant correlation with the yellow module ($\rho = 0.55$, $P = 6 \times 10^{-10}$, Supplementary Fig. 4e) among modules. Each type of the WHO classification corresponded to different modules, respectively; i.e., A, AB— black, turquoise; B1, B2 - blue; B2, B3 - yellow; C—green, red. The gray module was strongly associated with gender, and the black and turquoise modules with age at diagnosis. On the PCA plot based on the transcriptome, the enrichment of each module was well-coordinated with the profile of WHO types, suggesting that the heterogeneity of thymoma can be represented by the gene modules (Fig. 1c). Next, we investigated the detailed gene profiles of the modules associated with MG. In accordance with that MG was particularly enriched in the WHO type B epidemiologically[20] and in TCGA samples (Supplementary Fig. 4a), the yellow module linked to MG was associated with type B2 and B3 (Fig. 1c). In contrast, the blue module, which was independent of MG, was also associated with types B1 and B2 (Fig. 1c). To distinguish the yellow module from the blue one, we selected cytokeratins, which have various isotypes specific to tissues. Within cytokeratin isotypes, *KRT6A*, *KRT6C*, *KRT15* were specific to the yellow module, whereas *KRT7*, *KRT17*, *KRT18* were to the blue module (Fig. 1d). To confirm the difference histologically, we stained KRT6 and KRT17 proteins in thymoma tissue sections and observed the corresponding staining patterns to MG and non-MG-thymoma, respectively (Fig. 1e, f). We next examined the enriched pathways in the yellow module. Intriguingly, when we examined the enriched pathways in the yellow module, the most significantly enriched pathway was neuronal systems which included GABA receptors (*GABRA5*, *GABRB3*), neurofilaments (*NEFM*, *NEFL*), voltage-gated potassium channels (*KCNC1*, *KCNH2*, *KCNH5*, *KCND3*), and an NMDA receptor (*GRIN2A*) (Fig. 1g, h, Supplementary Data 3). Formation of the cornified envelope including cytokeratins and ion channel transport were also enriched in the yellow module. On the other hand, the other modules did not show the enrichment in neuronal systems but instead showed the enrichment of different types of pathways, such as interleukin-4 and interleukin-13 signaling in the blue module, extracellular matrix organization in

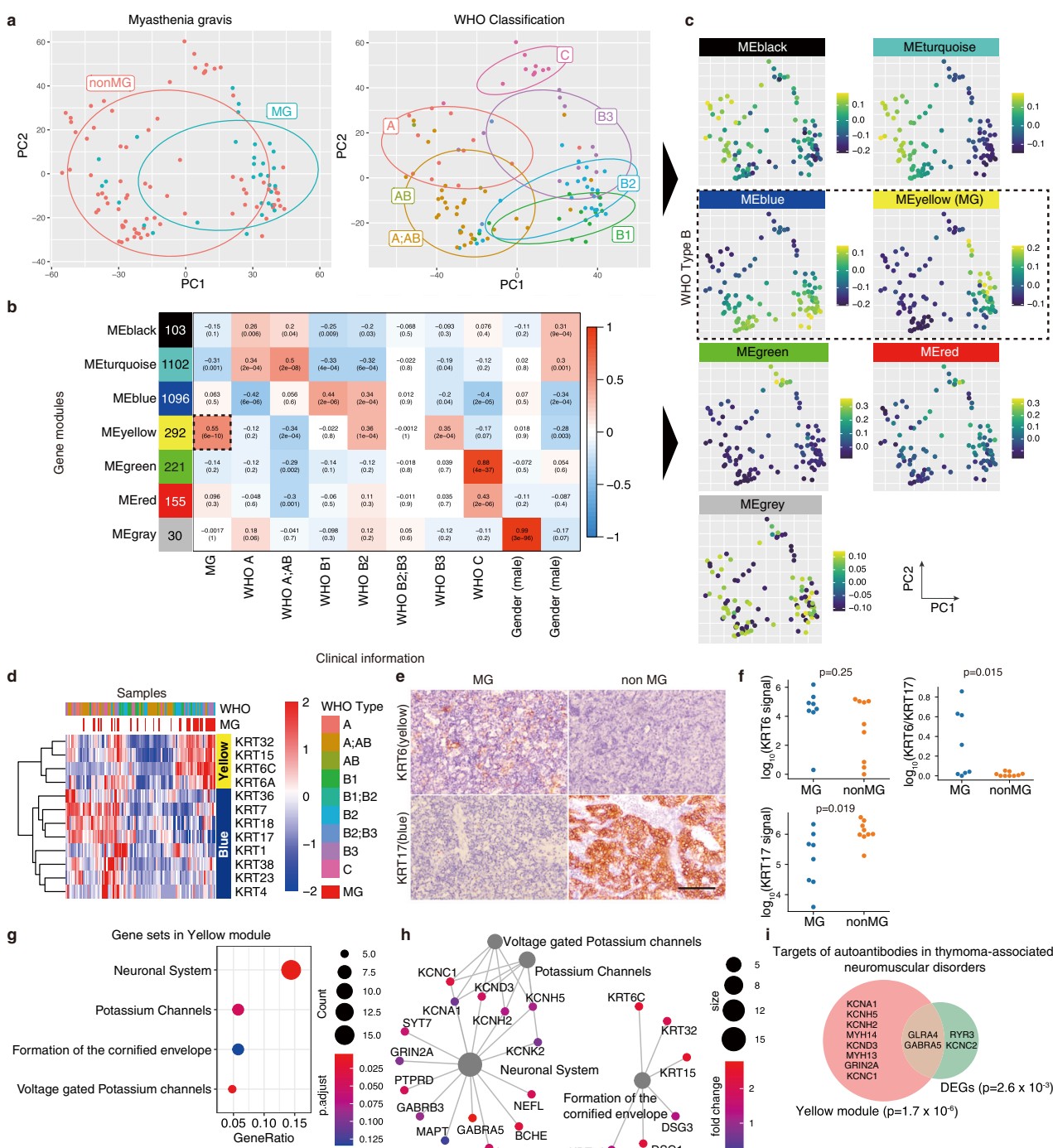

**Fig. 1 Transcriptome profiling of thymoma and MG-specific expression of neuro-related genes. a** PCA plots for transcription profile of thymomas from 116 patients. The left panel shows the disease status, MG or non-MG, and the right panel shows WHO classification based on histology. **b** Gene modules defined using WGCNA and the association with MG, WHO classification, gender, and age at diagnosis. Numbers in colored boxes on the left are the number of genes included in each module. The numbers in the heatmap show the correlation (upper) and the *P*-value (lower). **c** Eigengenes of each module for each patient on the PCA plot. **d** Heatmap of the gene expression of keratins in the yellow and blue modules. The color represents the Z-score of normalized expression by DESeq2. WHO classification and MG status were shown at the top of the heatmap. **e** Immunohistochemical (IHC) staining of KRT6 and KRT17 in MG and non-MG-thymoma. The scale bar: 100 μm. **f** Protein levels of KRT6 and KR17, and KRT6 normalized by KRT17 in MG and non-MG-thymoma quantified using microscopic images (MG $n = 8$, nonMG $n = 9$, Details are provided in Supplementary Fig. 2, 3 and Methods. Source data are provided as a Source Data file.). The signals were analyzed using a two-sided Mann–Whitney *U*-test. **g** Significantly enriched REACTOME pathways in the yellow module. The node size represents the number of genes included in each pathway, and the color represents the adjusted *P*-value of the enrichment. The pathways were sorted by the ratio of genes included in the yellow module. **h** Genes in enriched REACTOME pathway in the yellow module. Genes with *log₂ fold change* > 1 in comparison of MG and non-MG were selected. **i** Venn diagram showing overlap of targets of autoantibodies associated with thymoma with genes in the yellow module and upregulated genes in MG. Data were analyzed using a two-sided Fisher's exact test.

the green module, and interleukin-10 signaling in the red module (Supplementary Fig. 4f, Supplementary Data 4).

Thymoma has been shown to associate with paraneoplastic neurological diseases, such as encephalitis and myositis, besides myasthenia gravis[3,23]. We observed the significant overlap of candidate target antigens of thymoma-relating autoantibodies (Supplementary Data 6) with the yellow module genes (odds ratio = 7.87, $P = 1.65 \times 10^{-6}$) and more weakly with the differentially expressed genes between MG and non-MG (odds ratio = 7.42, $P = 2.67 \times 10^{-3}$). The overlap included an NMDA receptor (GRIN2), voltage-gated potassium channels (KCNH2, KCNC1, KCNA1, KCNC2, KCND3, KCNH5), a glycine receptor (GLRA4), a GABA receptor (GABRA5), and a ryanodine receptor (RYR3) (Fig. 1i).

We also examined immune profiles of MG, such as T-cell receptor (TCR)/B cell receptor (BCR) diversity and viral infections. The diversity of immunoglobulins in MG-thymoma was lower than that of non-MG-thymoma, suggesting that B cell maturation and expansion were occurred in MG-thymoma (Supplementary Fig. 5a). The diversity of TCR represented by the Gini index was mostly unchanged between them (Supplementary Fig. 5a), but the composition rate of a TCR alpha chain J, TRAJ24, was high in MG ($P_{adj} = 9.6 \times 10^{-4}$; Supplementary Fig. 5b), and especially the TRAJ24-TRAV13-2 combination was 7.50-fold more frequent in MG-thymoma (Supplementary Fig. 5c). To assess the effect of HLA on MG susceptibility, we determined major alleles of the HLA class I and II in MG using the same TCGA bulk RNA-seq dataset. The strongest association was observed in DQA1*01:04 (odds ratio = 4.43, $P = 0.050$), followed by DQB1*05:03 (odds ratio = 4.25, $p = 0.056$), A*24:02 (odds ratio = 2.84, $P = 0.058$, also reported by Machens et al.[24]) though all associations were below the significance level (Supplementary Fig. 5d). MG development has also been shown to associate with viral infections, including SARS-CoV2[25] and Epstein-Barr virus[26,27]. Therefore, we next examined the presence of viral transcripts in thymoma using the TCGA dataset. Although various viral transcripts such as Epstein-Barr virus and herpesvirus 6A were detected in MG-thymoma, no significant association with viruses was observed (False Discovery Rate (FDR) < 0.1, Supplementary Fig. 5e). In addition, we could not find any significant somatic mutations associated with MG, whereas missense mutations in GTF2I were observed in 49% of thymoma patients as previously reported[28] (Supplementary Fig. 6). Altogether, the unbiased large-scale omics analysis made visible MG-specific expression of neuromuscular molecules and the distortion in the diversity of TCRs and BCRs.

**Single-cell profiling of thymoma and PBMCs from anti-AChR antibody positive patients.** To clarify the source of neurological molecules and the surrounding immune environments in MG-type thymoma, we conducted single-cell RNA sequencing (scRNAseq) experiments of thymoma and peripheral blood mononuclear cells (PBMCs) derived from four anti-AChR antibody positive patients (Fig. 2a, Supplementary Fig. 7). The patients consisted of three females and one male, had not received immunosuppressive therapy preoperatively except for one patient, ranged in age from 35 to 55 years, and had thymoma type AB-B2 (Supplementary Data 7). Using a droplet-based single-cell isolation method, we profiled 33,839 cells from thymoma and 30,810 cells from PBMCs and identified 49 clusters upon them (Fig. 2b,c, Supplementary Fig. 8a, b). To analyze the similarity and differences between thymic cells and PBMCs, clustering was performed for the pooled cells. The cell annotation of PBMCs and the thymus was well-concordant with the previously reported scRNAseq experiments for healthy PBMCs[29] and the thymus[30]

(Supplementary Fig. 8d, e), and each cluster was well-separated by the specifically expressed genes. (Fig. 2d, Supplementary Fig. 8c, Supplementary Data 9, 10). In the latter parts, we analyzed the detailed expression profiles of the major clusters; stromal cells, T cells, and B cells, of MG-type thymoma.

**Identification of a unique thymic epithelial cell cluster in MG-type thymoma.** We profiled stromal cells of thymoma firstly. Clustered stromal cells corresponded to endothelial cells (positive for PECAM1/CD31, VWF), normal fibroblasts (FN1, EGFL6), tumor-associated fibroblasts (TAFs; PDGFRA, ADH1B), and thymic epithelial cells (TECs; KRT19, S100A14) (Fig. 3a, Supplementary Fig. 9a). We then extracted the TEC cluster and re-clustered them into cTEC (CCL25, PSMB11) and mTEC (CCL19, KRT7) clusters (Fig. 3b, c). The mTECs further fell into 3 clusters; mTEC(I) specifically expressing KRT15 and IFI27; mTEC(II) expressing CLDN4 and KRT7; and the unique mTECs expressing neuromuscular-related molecules (Fig. 3d). Cells in this unique mTEC cluster also expressed brain-specific genes included in the yellow module, such as GABRA5, MAP2, NEFL, NEFM, SOX15, TF. Their ectopic expression was also confirmed immunohistochemically in MG-thymoma tissue sections (Fig. 3e, Supplementary Fig. 9c, d). Among the yellow module genes, we selected KRT6 and GABRA5 as marker genes of nmTECs in the following criteria; (1) the expression was increased in MG patients in TCGA bulk RNA-seq dataset, (2) in the scRNAseq dataset, stably and preferentially expressed in nmTECs (Supplementary Fig. 9b). By immunohistochemistry (IHC), GABRA5 as one of the neuronal molecules expressed in the unique cluster and the cytokeratin KRT6, which belongs to the yellow module, were detected in identical cells (odds ratio = 50.6, $P < 10^{-16}$), with the cytoplasm and the pericellular localization of the cells, respectively (Fig. 3g, h, Supplementary Fig. 10). Due to the atypical expression profile of the cluster, we named the population neuromuscular-mTECs or nmTECs. nmTECs also expressed some of the targets of autoantibodies in thymoma-associated neuromuscular disorders highlighted in TCGA bulk RNA-seq analysis in Fig. 1i (Supplementary Fig. 9e). To assess the counterparts in the normal thymus, we compared scRNAseq data of thymoma TECs with that of the normal thymus previously published[30]. Thymoma nmTECs were partially correlated with an immature TEC cluster (mcTECs) and not with myoid cells (TEC(myo)) and neuroendocrine cells (TEC(neuro)) in the normal thymus (Fig. 3f). Next, to clarify the biological characteristics of nmTECs, gene set enrichment analysis of nmTECs was performed using the REACTOME gene sets. nmTECs showed the enhancement of pathways such as TP53 activation and pathways in cancer (Fig. 3i, k, Supplementary Data 11). nmTECs showed the highest number of detected reads per cell, while markers for other cell-types such as T cells and B cells were not detected in their expression (Fig. 3m, Supplementary Fig. 9f), suggesting that nmTECs are tumorous cells and not doublets. nmTECs also showed the enrichment in E3 ubiquitin ligases, IFNγ signaling, IFNα signaling, and class I MHC mediated antigen processing and presentation (Fig. 3i, j). Although MHC class II antigen presentation was not significantly enriched in the pathway analysis, nmTECs showed upregulation of HLA class II molecules and IFNγ together with downstream molecules of IFNγ signaling; STAT1, IRF1, CIITA, which have been shown to activate MHC class II regulations[31,32]. It suggests that nmTECs may possess a high capability of antigen-presentation via HLA class II (Fig. 3n). AIRE and FEZF2, which have been reported to be involved in the production of self-antigens in mTECs, and tissue-restricted antigens (TRAs) were expressed in a few cells of mTEC(I) cells, but not in nmTECs (Supplementary Fig. 9g). These observations

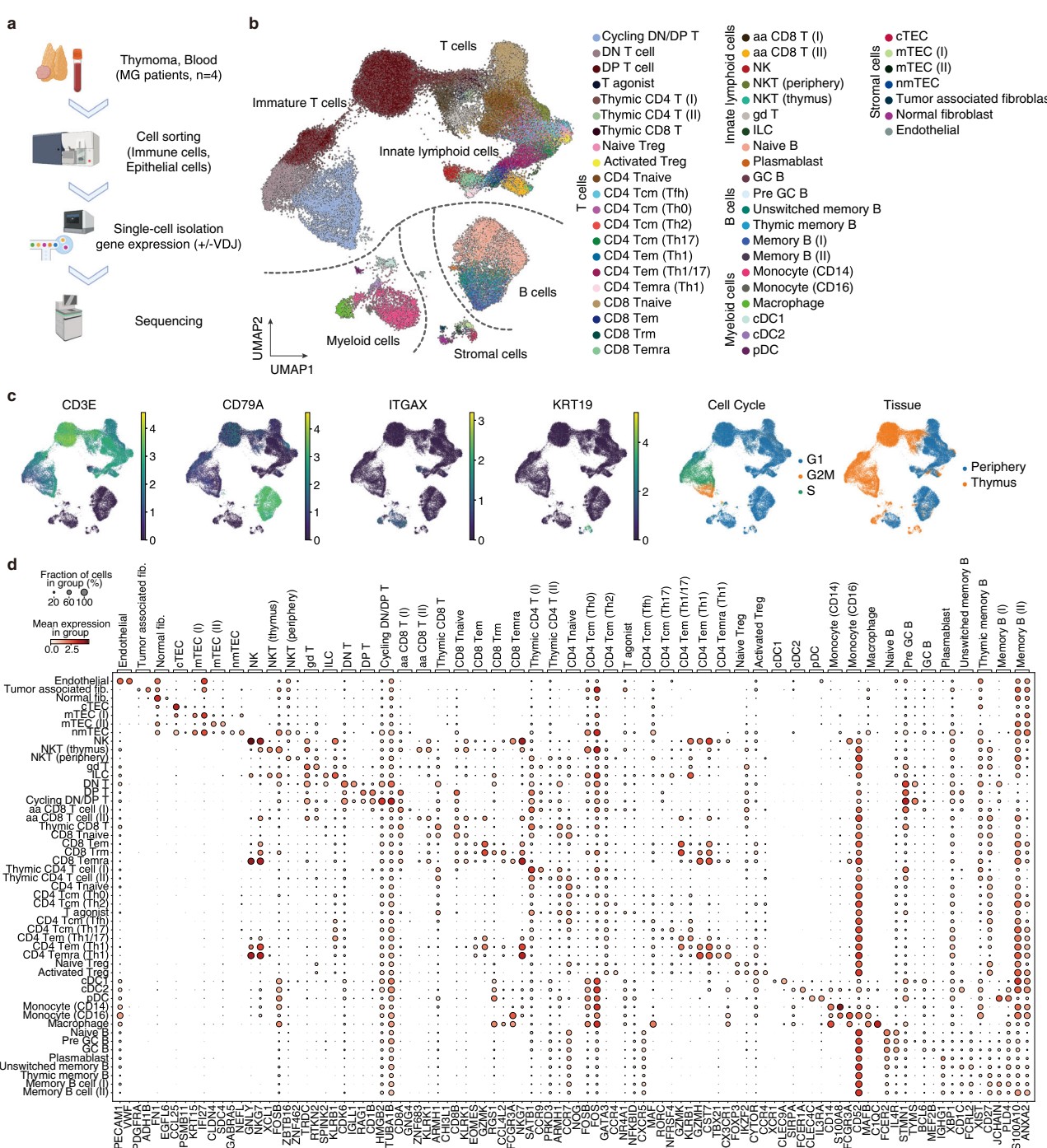

**Fig. 2 Overview of scRNAseq of thymoma and blood from anti-AChR receptor antibody positive patients. a** The experimental design of scRNAseq. Immune cells and non-immune cells from MG-type thymoma and immune cells from the blood of corresponding patients were collected for scRNAseq. **b** UMAP plot for 65,935 cells displaying the 49 clusters from thymoma and blood of MG patients. **c** UMAP plot of marker genes, inferred cell cycle, and tissue origins. **d** Dot plot depicting signature genes' mean expression levels and percentage of cells expressing them across clusters. The detailed dot plot is shown in Supplementary Fig. 8c.

thus suggest that nmTECs is a unique population producing neuromuscular-related molecules with active antigen presentation via MHC class I and II molecules.

**Dynamics of myeloid cells in MG-type thymoma.** To explore the MG-specific immune environment in thymoma, we next profiled myeloid cells. We identified six myeloid clusters in thymoma and PBMCs (Supplementary Fig. 11a, b). Monocytes were dominated in PBMCs, while macrophages and dendritic cells

were populated mostly in thymoma (Supplementary Fig. 11e). Among clusters, type 2 conventional dendritic cells or cDC2s (*CLEC10A*, *FCER1A*, *ITGAX*/CD11c), which preferentially polarize toward $T_H2$, $T_H17$, and $T_{FH}$ responses[33,34], were inferred to migrate from the periphery into thymoma from RNA velocity[35] (Supplementary Fig. 11c–f).

**B cell maturation with ectopic germinal center formation in MG-type thymoma.** Since B cells are the source of the

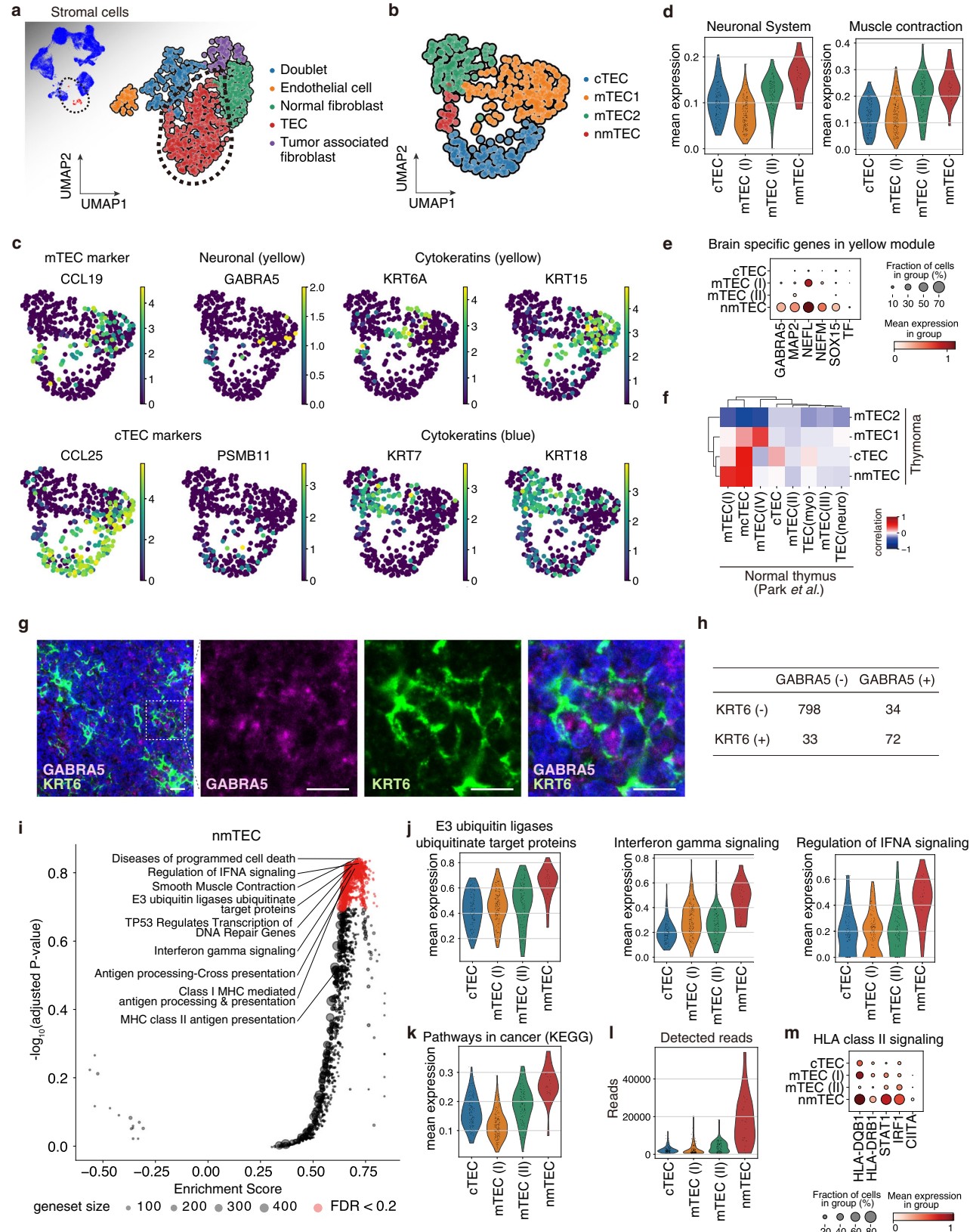

autoantibodies causative for MG, we next assessed B cell dynamics in MG-type thymoma. To determine the subpopulations of B cells, we categorized them into eight distinct B cell clusters. Notably, we found a population forming a germinal center (GC; positive for *BCL6, MEF2B*) in MG-type thymoma (Fig. 4a), while GC B cells were not detected in the normal thymus (Supplementary Fig. 12d). The formation of ectopic germinal centers in MG-thymoma was also histologically confirmed (Fig. 4f). Based on the expression of immunoglobulins, B cells were divided into three groups; (1) Naive, GC, pre-GC (*IGHM, IGHD, IGHG3* high);(2) memory B cells (*IGHA1, IGHA2, IGHG2* high); and (3) plasmablasts (*IGHG1, IGHG3,*

**Fig. 3 Neuromuscular thymic epithelial cells (nmTECs) expressed neuromuscular genes, IFN gamma signaling pathway genes, and HLA molecules.**
**a**, **b** UMAP embedding for stromal clusters (**a**) and thymic epithelial cells (TECs) clusters (**b**) in thymoma. **c** Gene expression of marker genes on UMAP embedding. d, Violin plots of mean expression of the REACTOME gene sets; Neuronal System (left) and Muscle contraction (right) in TEC clusters. **e** Dot plot of the yellow module genes. Corresponding protein expressions were also confirmed using IHC (Supplementary Fig. 9b). **f** Heatmap showing correlation of transcriptional profile with TEC cells in thymoma (this publication) and a normal thymus (Park et al.[30]). **g** Immunofluorescence staining for confirming the presence of nmTECs positive for GABRA5 (red), KRT6 (green), and DAPI (blue). Scale bars: 20 μm. **h** Cross table showing cell numbers of GABRA5 positive/negative and KRT6 positive/negative cells in a representative IHC slide. Data were analyzed using a two-sided Fisher's exact test.
**i** Volcano plot showing REACTOME gene sets enriched in nmTECs. **j–l** Violin plots of mean expression of the gene sets (**j–k**) and the number of detected reads per cell (**l**). **j** Represents the significantly enriched REACTOME gene sets and **k** represents the KEGG gene set, Pathways in cancer. **m** Dot plot of gene expression of HLA class II-related molecules in TECs.

*IGHG4* high) (Fig. 4b). We also observed that a pre-GC B cell population (*STMN1, TCL1A*) was preferentially enriched in thymoma (Fig. 4c). The RNA velocity analysis showed that pre-GC cells were directed from naive B cells toward GC B cells, memory B cells, and plasmablasts in MG-type thymoma (Supplementary Fig. 12a), suggesting that the B cell maturation progresses normally in the MG-type thymoma. In addition, Pre-GC, GC, thymic memory B cells, and plasmablasts were enriched in thymoma compared to PBMCs (Fig. 4d, e).

**T-cell polarization in MG-type thymoma.** In the thymus, T cells are characteristically educated by antigen-presenting cells, including mTECs, and abnormalities of the antigen-presentation frequently associate with autoimmune diseases via T-cell dysfunction[36,37]. Therefore, we next investigated whether T cells in thymoma were engaged in MG pathogenesis (Supplementary Fig. 12d). We observed the existence of immature and mature T cells in thymoma, suggesting that the physiological T-cell development was maintained even in MG-type thymoma. Among populations, we identified a thymoma-specific mature T-cell population, CD8$^+$ tissue-resident memory T cell (CD8 T$_{RM}$) expressing *CXCR6* as seen in other tissues such as the lung[38,39] and skin[40] (Supplementary Fig. 12d). We next focused on CD4$^+$ T-cell clusters, which are essential for B cell activation. In thymoma and PBMCs, we identified 13 specific clusters, which corresponded to cells in the process of differentiation, i.e., immature thymic CD4$^+$ T cells to terminally differentiated effector memory CD4$^+$ T cells (CD4 T$_{EMRA}$). T cells after thymic selection contained CD4$^+$ naive T cells (CD4 T$_{NAIVE}$; *CCR7$^+$ FAS$^-$*), CD4$^+$ central memory T cells (CD4 T$_{CM}$; *CCR7$^+$ FAS$^+$*), effector memory T cells (CD4 T$_{EM}$; *CCR7$^-$ FAS$^+$*), and terminally differentiated effector memory CD4$^+$ T cells (CD4 T$_{EMRA}$; *FAS$^+$ CD28$^-$*) (Fig. 4g, h). We also identified T-cell polarizations using characteristic transcription factors and chemokine receptors such as T$_H$1 (*TBX21*/Tbet) in T$_{EM}$ and T$_{EMRA}$, T$_H$2 (*GATA3, CCR4*) in T$_{CM}$, T$_H$17 (*RORC, CCR6*) in T$_{CM}$, T follicular helper cells (T$_{FH}$; *CXCR5, PDCD1*) in T$_{CM}$ (Fig. 4h). These cell annotations were also concordant with the bulk RNA-seq dataset of purified T cells[41] (Supplementary Fig. 12e). When we assessed the tissue localization of these cells, CD4 T$_{CM}$ (T$_H$0) was more abundant in the thymus, and CD4 T$_{CM}$ (T$_{FH}$) was equally abundant in the thymus, whereas other memory T cells such as CD4 T$_{CM}$ (T$_H$2), CD4 T$_{CM}$ (T$_H$17) were significantly more abundant in the periphery, suggesting that T$_H$0-T$_{FH}$ axis are prominent in the thymus (Fig. 4i, j). Next, to infer T-cell dynamics between thymoma and periphery, we investigated the commonalities of T-cell receptor (TCR) repertoires of these cell populations. Strong clonal expansions were observed in T$_H$1 prone clusters, including CD4 T$_{EM}$ (T$_H$1/17), CD4 T$_{EM}$ (T$_H$1) and T$_{EMRA}$ (T$_H$1), and also slight clonal expansions in CD4 T$_{CM}$ (T$_H$2), CD4 T$_{CM}$ (T$_H$17), and activated T$_{reg}$ cells (Supplementary Fig. 12f). By examining TCR similarity between the thymus and periphery for each cluster, T$_{reg}$ cells showed higher levels of TCR similarity between the thymus

and the periphery, compared to the other cell populations (Fig. 4l). This suggests that naive T$_{reg}$ cells are activated in thymoma aberrantly and circulated into the periphery. In addition, a chemokine receptor *CXCR4* was preferentially expressed in thymic mature T cells (Fig. 4k, Supplementary Fig. 12g). A couple of thymic T cell-specific genes such as *CD69, SOCS1* (STAT-Induced STAT Inhibitor 1) and *RGS1* (Regulator Of G-Protein Signaling 1) were also expressed in B cells in thymoma, but not in periphery (Supplementary Fig. 12b, c), suggesting that these genes were regulated by a shared tissue-specific program between T and B cells. Taken together, a detailed analysis of B and T cells revealed that MG-type thymoma kept primary lymphoid tissue characteristics for T-cell education and gained abnormal inflammatory profiles with ectopic GC formations.

**Cell–cell interaction inference.** To analyze the communications among cells, we next inferred cell–cell interaction by integrating single-cell data with a curated ligand-receptor pair database through a bioinformatics application, *CellPhoneDB*[42] (Fig. 5a). The cell fraction possessing the highest number of intercellular interactions was nmTECs (Fig. 5b). The cell–cell interaction network analysis showed that nmTECs acted as a hub in the network and interacted with myeloid cells, T cells, B cells, tumor-associated fibroblasts, and endothelial cells (Fig. 5c). nmTECs and tumor-associated fibroblasts preferentially expressed *CXCL12*, and thymic B cells and helper T cells, including T$_{FH}$ and T$_{reg}$ cells, expressed its receptor, *CXCR4*. Given that the *CXCR4-CXCL12* axis has been shown to be important in T-cell homing in synovial tissues of rheumatoid arthritis[43], neurogenesis[44], and maintenance of hematopoietic stem cells[45], the interaction may be important for nmTEC-mediated T-cell regulation (Fig. 5d). *CXCR5* was expressed in B cells, T$_{FH}$, and CD8 T$_{RM}$, while its ligand, *CXCL13*, was expressed in T$_{FH}$ and CD8 T$_{RM}$, suggesting that the putative role of *CXCR5-CXCL13* for T-B interaction in thymoma. This predicted interaction was consistent with the previous findings[46]. We also predicted the interactions of nmTECs with vascular endothelial cells via *VEGFA* and *VEGFE* and with tumor-associated fibroblasts via *PDGFA-PDGFRA*. To verify the predicted interaction, we performed immunostaining of CD31 on MG-type thymoma sections and observed that GABRA5$^+$ nmTECs were in proximity to CD31$^+$ vascular endothelial cells (Fig. 5e–g, Supplementary Fig. 13). These observations suggest that nmTECs may promote angiogenesis via the interaction of vascular endothelial cells, which is consistent with previous reports[17].

**Integrative analysis of MG pathology across cell types.** Recently, a computational method has been developed to infer the cell proportions from bulk RNA-seq datasets using references constituted by scRNAseq[47]. To identify cell populations enriched in MG, we estimated cell distribution by deconvolution of large-scale bulk RNA-seq of thymomas in the TCGA database, using

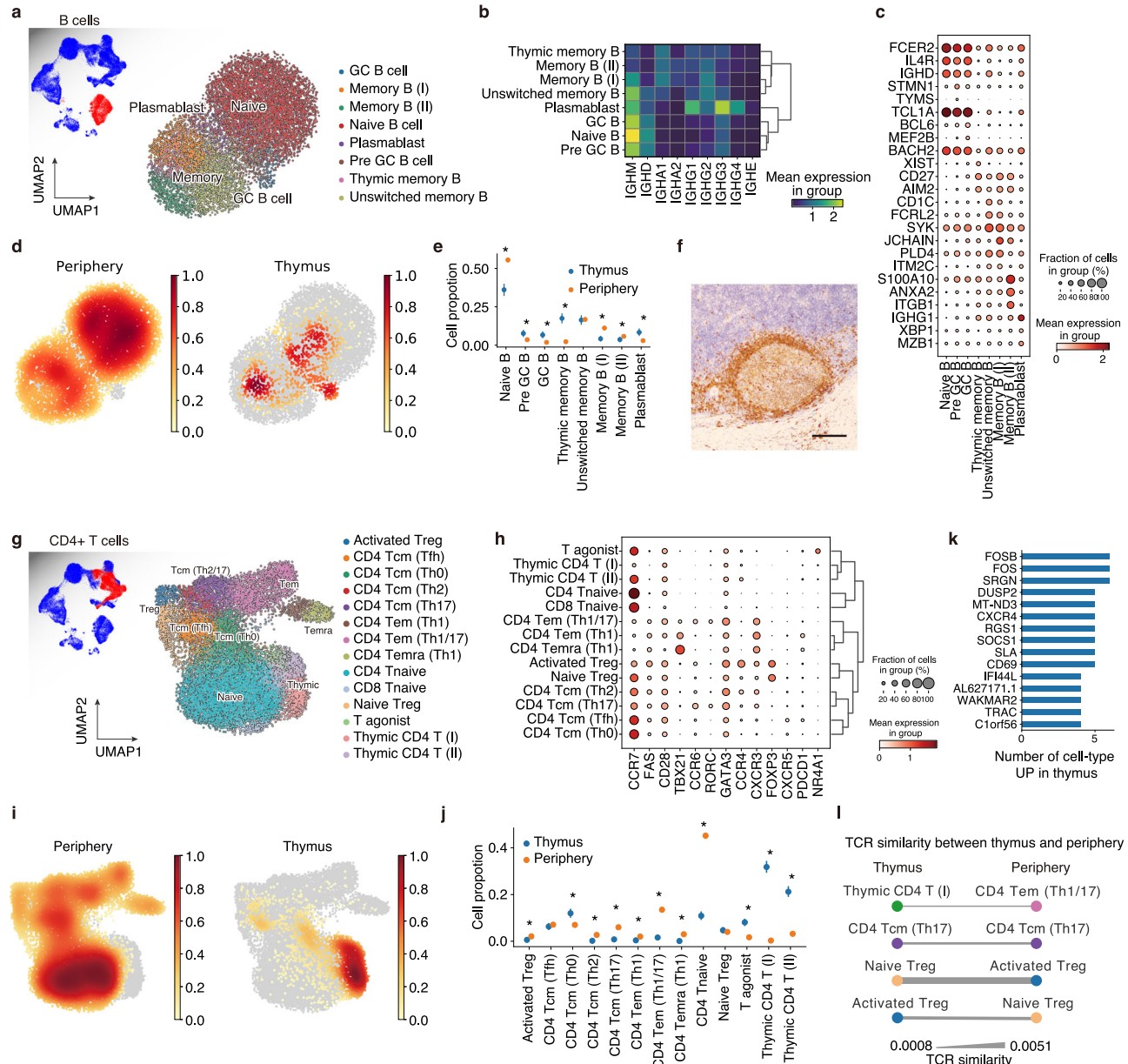

**Fig. 4 Immune cell landscape elucidates GC formation, $T_HO$-$T_{FH}$ enhancement, and Treg recirculation in MG-type thymoma. a** UMAP embedding for B cell clusters of thymoma and peripheral blood. **b** Heatmap of immunoglobulin expressions in each B cell cluster. Mean expressions in each group are shown as a heatmap. **c** Dot plot of gene expression of marker genes of each B cell cluster. **d** Density plots showing B cell accumulation in the periphery (left) and thymus (right). **e** Cell proportion of each B cell cluster in thymoma and peripheral blood. Periphery $n = 2$, Thymoma $n = 4$. Error bars show 98% highest density interval. *$FDR < 0.05$. (Methods) **f** Representative IHC image of germinal center in MG-thymoma stained for CD79A. Scale bar: 100 μm. The presence of GC was also checked in Fig. 6j and Source Data. **g** UMAP embedding for CD4+ T-cell clusters of thymoma and peripheral blood. **h** Dot plot of gene expression of marker genes of each T-cell cluster. **i** Density plots showing T-cell accumulation in the periphery (left) and thymus (right). **j** Cell proportion of each T-cell cluster in thymoma and peripheral blood. Periphery $n = 2$, Thymoma $n = 4$. Error bars show 98% highest density interval. Center points represent the mean of the posterior distributions. *$FDR < 0.05$. (Methods) **k** Bar plot of thymus specific genes across CD4+ T-cell clusters ranked by the number of cell types where each gene was upregulated ($P_{adj} < 0.05$ and $log_2$ fold change $> 1$) in mature CD4+ T cells. **l** TCR similarity between peripheral blood and thymoma. The thicknesses of edges represents TCR similarity. *$FDR < 0.05$ in **e** and **j**. Statical procedures in **e** and **j** are described in Methods. GC germinal center.

detailed single-cell annotation defined in the previous sections. Among cell populations, cTECs were accumulated in WHO type A; mTEC(I) in type A, B3, C; mTEC(II) in type A; and immature T cells in type B1 thymoma (Supplementary Fig. 14a). These observations were concordant with the phenotypes defined by WHO classification, suggesting that the deconvolution was functioning well. The numbers of cycling DN/DP T cells and endothelial cells were decreased and increased, respectively, along with age (Supplementary Fig. 14b). The most significantly associated cell population to MG was nmTECs, followed by GC B cells and cDC2s (Fig. 6a–c).

Next, we examined the contribution of each cell type to each module defined by WGCNA. nmTECs were the most significantly contributed to the yellow module, which was associated

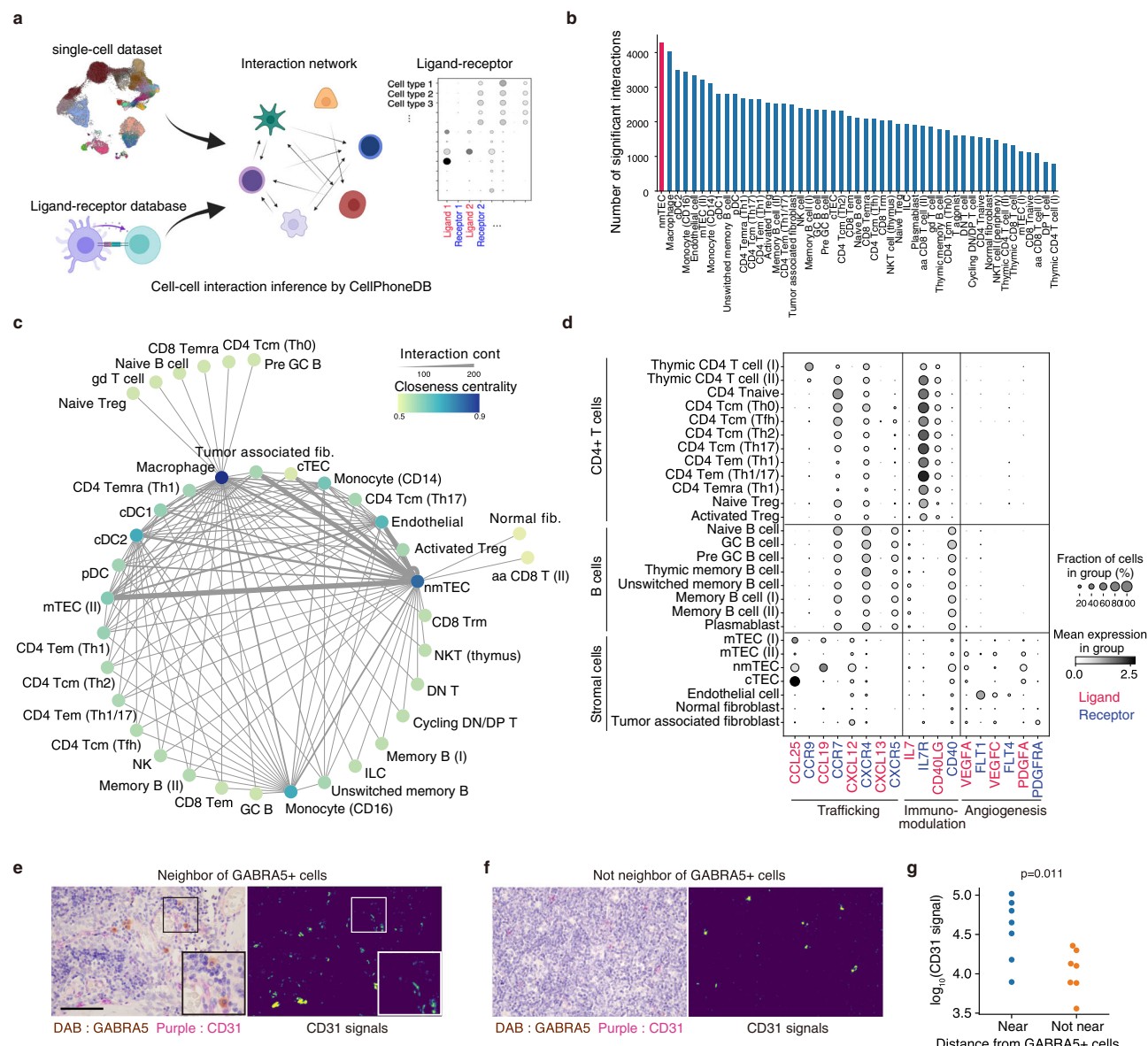

**Fig. 5 nmTECs strongly associated with epithelial cells, myeloid cells, and T cells with characteristic ligand-receptor pairs. a** Schematic view of the cell–cell interaction analysis. **b** Bar chart showing the number of significant interactions with other cell types in each cell type. **c** Cell–cell interaction network inferred from scRNAseq data. Each node represents a cell type, and the thickness of each edge represents the number of significant interactions. Edges with <75 significant interactions were removed. **d** Dot plot of gene expression of ligand-receptor pairs involved in trafficking, immunomodulation, and angiogenesis in CD4+ T cells, B cells, and stromal cells. **e, f** Representative images of the colocalization of nmTECs (GABRA5; DAB) and endothelial cells (CD31; purple) in the vicinity of GABRA5+ cells (**e**) and not in the vicinity of GABRA5+ cells (**f**). Binarized signals are shown in yellow. Scale bar: 100 μm. **g** Protein levels of CD31 near and not near from GABRA5+ cells in MG-thymoma quantified using microscopic images (Near $n = 7$, Non-Near $n = 7$. Details are provided in Supplementary Fig. 13 and Methods. Source data are also provided as a Source Data file.). For each group, seven areas from four MG patients were quantified. The signals were analyzed using a two-sided Mann–Whitney U-test.

with MG (mean expression = 0.092, $P_{adj} < 10^{-13}$; Fig. 6d, Supplementary Data 17). The blue module, which was also associated with WHO type B, was found to be associated with cycling DN/DP T cells and DP cells (mean expression = 0.092 and = 0.10, $P_{adj} < 10^{-100}$ and $<10^{-100}$; Fig. 6d, Supplementary Data 17). The target molecules of thymoma-associated autoantibodies were also enriched in nmTECs significantly (mean expression = 0.014, $P_{adj} = 1.4 \times 10^{-3}$; Supplementary Fig. 16c, Supplementary Data 17). To measure genetical effects on each cell population, we listed up myasthenia gravis associated genes reported in three genome-wide association studies (Seldin et al.[48], 532 cases, 2128 controls; Renton et al.[49], 1455 cases, 2465 controls; Gregersen

et al.[50], 649 cases, 2596 controls, Supplementary Data 16). We excluded HLA genes, which possessed the most significant signals, to avoid the ambiguity derived from the complex linkage disequilibrium (LD) structure in the HLA regions. We also extracted genes associated with genome-wide association studies (GWAS) SNPs in consideration of expression quantitative trait locus (eQTL) and LD structures (see Methods). MG-associated genes in both lists were significantly associated with $T_{reg}$ cells and B cells, including GC B cells and plasmablasts (Fig. 6d, Supplementary Fig. 16a, b, Supplementary Data 17). Overall, these analyses indicated that nmTECs, GC B cells, and cDC2s were atypically increased in MG-thymoma and that the genetic

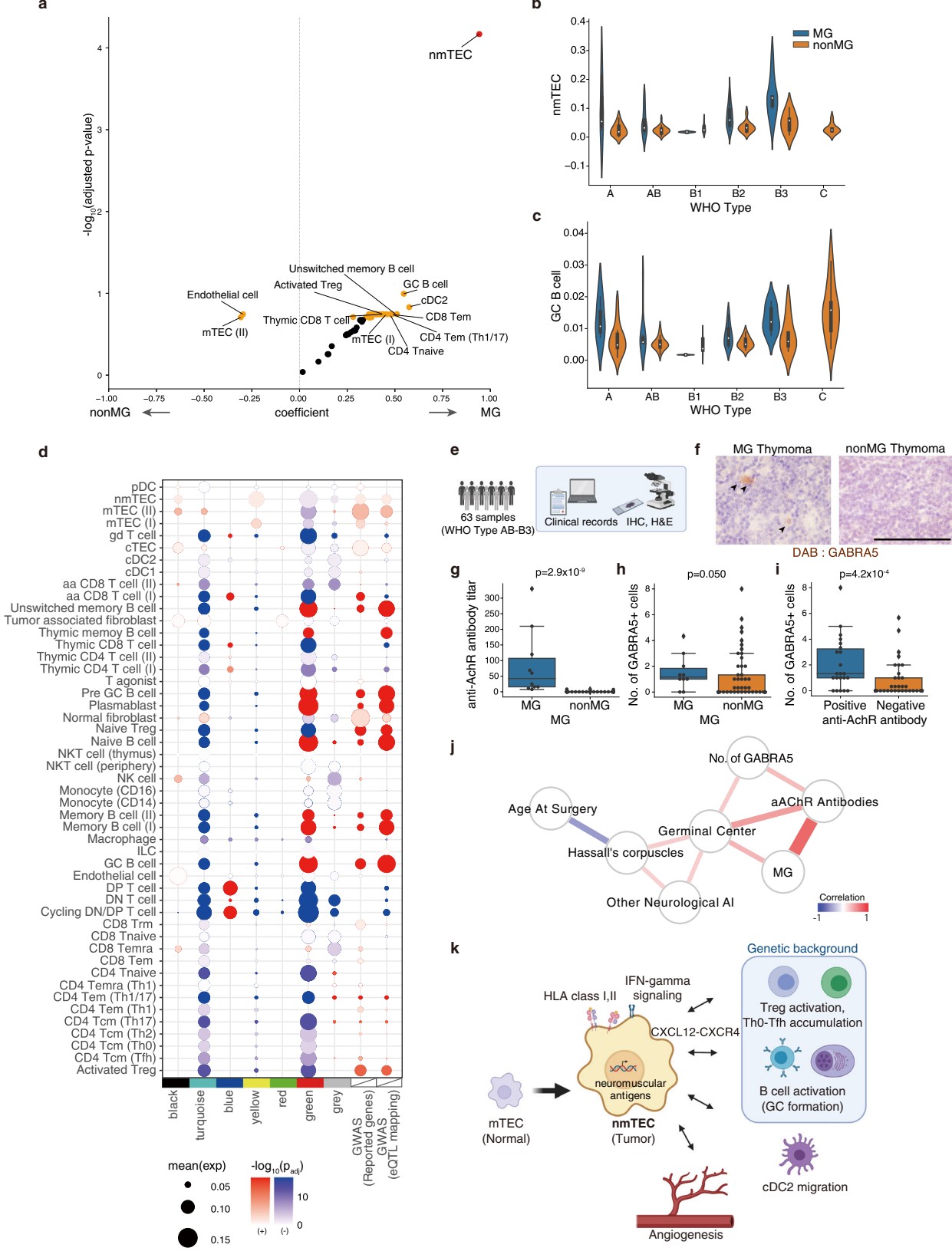

effects associated with MG were mainly accumulated in T and B cells.

**Histological validation of the MG-associated phenotypes.** To validate the MG-associated changes in another cohort, we examined tissue specimens from 63 WHO type AB-B3 thymoma

surgery cases with the clinical records (Fig. 6e). To quantify the amount of nmTECs, we performed immunostaining for GABRA5 (Fig. 6f) and found that the number of GABRA5-positive cells was higher in MG ($P = 0.050$) and more significantly in anti-AChR antibody-positive thymoma patients ($P = 4.2 \times 10^{-4}$, Fig. 6g–i). In addition, the presence of germinal centers

**Fig. 6 Cell-type wide analysis exhibits nmTECs and GC B cells associate with MG. a** Volcano plot showing the association with MG for deconvoluted cell proportions s, which were calculated from TCGA bulk RNA-seq dataset ($n = 116$) with the reference defined in our scRNAseq analysis. Coefficients and p-values were calculated with multiple regression (Methods). Red dot $FDR < 0.05$, orange dots $FDR < 0.2$. **b, c** Violin plots of the inferred cell proportion of TCGA bulk RNA-seq dataset ($n = 116$) for nmTECs (**b**) and GC B cells (**c**) partitioned by WHO classification and MG status. Box plots show IQRs and whiskers show the maximum or minimum value in the dataset excluding outliers ($Q3 + 1.5 \times IQR$ or $Q1 - 1.5 \times IQR$). **d** Expression enrichment of gene modules, targets of autoantibodies in thymoma-associated neuromuscular disorders, and GWAS-reported genes for early-onset MG (EOMG) and late-onset MG (LOMG). The enrichment score for each gene set was analyzed using a two-sided Mann–Whitney $U$-test across cell-type, and the adjusted $P$-value was calculated. A positive correlation is colored in red, and a negative correlation is in blue. **e** Strategy of histological assessment by an independent cohort. **f** Representative Immunohistochemical (IHC) staining images of GABRA5 in MG (left) and non-MG (right) thymoma. Arrowheads indicate GABRA5-positive cells. Scale bar: 100 μm. **g–i** Box plots of anti-AChR antibody titer (nmol/L) (**g**) and the number of GABRA5-positive cells in thymoma (**h**) in MG and non-MG-thymoma patients, and the number of GABRA5-positive cells in thymoma partitioned by anti-AChR antibody titer (**i**). Data were analyzed using a two-sided Mann–Whitney $U$-test. Box plots show IQRs and whiskers show the maximum or minimum value in the dataset excluding outliers ($Q3 + 1.5 \times IQR$ or $Q1 - 1.5 \times IQR$). Source data are provided as a Source Data file. **j** Network showing the correlation with clinical and histological features. Anti-AChR antibody titer was tested before the thymectomy. The existence of the germinal center was determined using H&E staining or DAB staining for CD79A. Statistically significant edges with the multiple test correction were retained ($FDR < 0.2$). The edge color represents Pearson's correlation, and the thickness of the edge represents $-log_{10}FDR$. Source data are provided as a Source Data file. **k** Proposed MG pathology in thymoma.

determined by H&E staining was associated with the increase of anti-AChR antibodies, the presence of MG/other neuro-related autoimmune diseases, and the number of GABRA5-positive cells (Fig. 6j). These observations depicted that the emergence of nmTECs was involved in MG pathogenesis in thymoma together with the altered immune cell populations.

## Discussion

In this study, we showed the pathogenic changes responsible for MG in thymoma by exploring MG-deviated expression at the single-cell level. As a key finding, we identified abnormal expression of neuromuscular molecules specific to MG cases within thymoma. Single-cell RNA-seq and immunohistological examination of MG-type thymoma specimens revealed that these neuromuscular expressions were limited in a subpopulation of mTECs (GABRA5+KRT6+), termed nmTECs. In addition, MG-type thymoma developed atypical immune microenvironments with GC formation, B cell maturation, and ectopic neuromuscular expression on nmTECs, providing a holistic picture of the cell dynamics for producing autoantibodies, which was previously known only in fragments (Fig. 6k).

While TAMG is caused by autoantibodies against acetylcholine receptors expressed at the neuromuscular junction under normal conditions, the mechanisms by which those autoantibodies are generated have not been clarified so far. In this study, integrated omics analysis showed that responsible antigen-presenting cells to present acetylcholine receptors would be nmTECs in the thymus. mTECs originally possess the ability to express systemic antigens ectopically using a transcription factor, *AIRE*, to eliminate self-reactive T cells[7]. In fact, it has been shown that mTECs acquire a variety of cell polarities such as tuft, keratinocyte-like, and neuroendocrine after *AIRE* expression[51,52]. Therefore, it seems likely that acetylcholine receptor expression by nmTECs would be caused by the intrinsic ability of mTECs to present self-antigens under the negative selection. The expression of autoantigens is also known to be enhanced by IFN-γ[53]. We observed that the IFN-γ signaling cascade in nmTECs was more active than those in normal mTECs, indicating that they present antigens to immune cells more efficiently. Thus, nmTECs would feed self-antigens to auto-reactive lymphocytes and trigger pathological GC formation in the thymoma. This also gives rise to the possibility that the physiological production of self-antigens by mTECs might have a risk of inducing autoimmunity. Interestingly, MG-thymoma expresses not only acetylcholine receptors but also various neuromuscular-related antigens associated with other autoimmune diseases, suggesting that the abnormal

expression of neuromuscular antigens by nmTECs is also associated with thymoma-associated neuromuscular autoimmune diseases. In addition, while the most frequent complication of thymoma is anti-acetylcholine receptor antibody-positive myasthenia gravis, the increase of the *CHRNA1* expression was milder than other neuromuscular antigens such as *GABRA5* and *RYR3*. The moderate increase of *CHRNA1* expression may be sufficient for causing serious symptoms due to the accessibility of acetylcholine receptor antibodies[54] and the importance of their biological functions[55]. Overall, these observations may provide clues to elucidate the pathogenesis of a wide range of neurological autoimmune diseases.

We have succeeded in capturing the entire picture of the thymic microenvironment for producing autoantibodies causative for MG. It is widely accepted that mature B cells in the thymus serve as a source of autoantibodies[56]. In addition, GC formation and an increase of $T_{FH}$ cells in the thymus have been reported as immune changes in MG-thymoma[17,18]. Our results were fully consistent with those observations and further revealed the accumulation of cDC2, which are considered as migrating DCs from the peripherally for supporting B cell maturation[57], in MG-type thymoma. Cell–cell interaction analysis also predicted that the *CXCR4-CXCL12*-mediated interaction between lymphocytes and nmTEC in the thymus is one of the key interactions for producing autoantibodies in the thymoma microenvironments in concordance with previous reports[58]. The interaction of nmTECs and lymphocytes together with cDC2, $T_{FH}$, GC accumulation suggests that there may be MG-specific immune microenvironments that support the maturation of autoantibody-producing B cells and their migration to the periphery. It has also been reported that anti-AChR antibody-producing cells reside in the bone marrow[59] and lymph nodes[60] outside the thymus and that antibody-producing cells continued to circulate in the periphery after thymectomy[61]. The circulation between the thymoma and the periphery also seems to be present in T cells, as suggested by our TCR repertoire analysis. We thus now have a better understanding of the thymoma microenvironment in which auto-reactive B cells are maturated with the help of neuromuscular molecule-presenting nmTECs, the construction of GC formation, enhanced Tfh cell activity, and cDC2 accumulation.

One of the remaining questions is whether the expression of neuromuscular molecules by mTECs triggers the MG development. Our data showed that some patients with high expression of neuromuscular genes did not develop MG (Fig. 1a, c). Regarding WHO classification, MG complications were observed not only in B3, which has a high epithelium, but also in B2 and B1 (Supplementary Fig. 4a). Histological analysis also showed

that GABRA5-positive, or nmTEC marker-positive, cells were present in some acetylcholine receptor antibody-negative patients (Fig. 6e–j). These results suggest that the accumulation of neuromuscular-related antigens induces a pre-disease state and is not a sufficient condition for MG pathogenesis. In other words, MG pathogenesis requires additional factors except for the differentiation of nmTECs. One of the candidate factors is viral infections since viral infections have been reported to be involved in many autoimmune diseases, including MG[25–27], via inducing immune disruption. While we could not detect any virus that significantly correlated with MG, its effect might contribute to the MG pathogenesis as reported. Another pathological factor is the genetic factors. The integrated analysis with GWAS reaffirmed the importance of T cells including $T_{reg}$ cells and B cells as a genetic predisposition for MG pathogenesis. Therefore, MG would be cooperatively developed by the expression of neuromuscular-related antigens, skewed immune microenvironment, genetic backgrounds, and environmental factors including virus infections. Further analysis will be required for addressing the stepwise development of MG.

Finally, we characterized the complex relationship between MG and thymoma from a view of cell composition and the source of neuromuscular molecules causative for MG. We hope that this study will provide useful information for the development of MG therapy.

## Methods

**Human samples**. The study using human samples was reviewed and approved by the Research Ethics Committee of Osaka University and carried out in accordance with the guidelines and regulations. Human samples were collected under approved Osaka University's review board protocols: ID 10038-9 and ID 850-2. Written informed consent was obtained from all donors.

**Immunohistochemistry**. All tissue samples were fixed in 10% formalin, embedded in paraffin, cut into 4-μm-thick sections. For DAB staining, Immunohistochemical staining was performed using the Roche BenchMark ULTRA IHC/ISH Staining Module (Ventana Medical Systems) with the Ultra CC1 mild protocol. For Double stains, we performed a second stain for slides that were DAB stained by the Ultra CC1 mild protocol using the Stayright Purple kit (AAT Bioquest). For multicolor fluorescent staining, we stained slides using Opal 4-Color IHC Kits (AKOYA Biosciences) and observed using the Zeiss LSM 710 or LSM 880 confocal microscope and ZEN microscope software (Carl Zeiss). The primary antigens and dilution ratios used are presented in Supplementary Data 18. The scoring of immunohistochemical staining images was supervised by the pathologists (K.K. and S.N.).

**Histological quantifications**. For DAB signal quantification, the region with the strongest DAB signal in each slide was captured. Double-stained slides of CD31 and GABRA5 were captured up to two GABRA5-positive areas and an equal number of negative areas from each MG-type thymoma specimen under 40x objective. After adjusting the white balance, signals of hematoxylin and DAB or hematoxylin, DAB, and Purple were separated using the reb2hed function in a python package scikit-image (v0.18.1) and quantified the areas above the threshold (Supplementary Figs. 2, 3, 13). For the distance between GABRA5 signals and KRT6 signals, the distances between the nearest blobs of GABRA5 and KRT6 were measured (Supplementary Fig. 10) were measured. Blobs were defined using the blob_log function provided by a python package scikit-image (0.18.1). The number of nmTECs in Fig. 6f was calculated by averaging the number of positive cells in the three region with the highest accumulation of positive cells under x40 objective using DAB staining for GABRA5. The detection of germinal centers was judged by H&E staining or IHC of CD79A. The existence of Hassall's corpuscles was judged by HE staining.

**Cell preparation and sequencing of scRNAseq**. We enrolled anti-AChR antibody positive patients for scRNAseq analysis (Supplementary Data 7). To ensure the quality of the library, the library preparation of all thymoma and peripheral blood samples was completed by the next day after the collection. Immune cells and thymic epithelial cells were isolated from thymic tissue dissected surgically, as previously described[62]. The sections were mainly consisted of thymoma with a small proportion of marginal regions. Briefly, thymic tissue was mechanically disrupted, and the fraction containing lymphocytes was collected. Extracted cells were stained with 7-AAD (BD Biosciences), and live cells were collected as a lymphocyte fraction. The remaining thymic tissue was subjected to enzymatic

treatment (Collagenase A (Worthington), DNAse I (Roche, Basel Switzerland), Trypsin/EDTA (nacalai tesque)) and the resulting cells were then subjected to a percoll density gradient centrifugation for the enrichment of thymic epithelial cells. Cells derived from low-density fraction were stained using FITC-labeled anti-EpCAM mAb (dilution: 1/10, HEA-125, Miltenyi Biotec), PE-labeled anti-CD45 mAb (dilution: 1/100, HI30, Biolegend). Dead cells were excluded by 7-AAD staining, and CD45 (low) EpCAM (high) was defined as thymic epithelial cells. Immune cells and thymic epithelial cells were isolated using BD Biosciences FACS Aria II. The gating strategy is described in Supplementary Fig. 7. To increase the collection of EPCAM+ cells, we used the lower threshold of EPCAM for the gating, resulting in the inclusion of fibroblasts and endothelial cells. For CD4+ T cells and B cells, we first collected PBMCs using Ficoll-Paque (Cytiva). Isolated PBMCs were washed, blocked Fc receptors using Fc Receptor Binding Inhibitor Polyclonal Antibody, Functional Grade, eBioscience™ (Thermo Fisher Scientific), and stained using FITC-labeled anti-CD3 mAb (dilution: 1/100, UCHT1, BD Bioscience), APC-labeled anti-CD4 mAb (dilution: 1/100, RPA-T4, Thermo Fisher Scientific), PE-labeled anti-CD19 mAb (HIB19, BioLegend), Live/Dead (Thermo Fisher Scientific). Then, live-CD3+CD4+CD19- cells and live-CD3-CD4-CD19+ cells were isolated using BD Biosciences FACS Aria II. FACS data analysis was performed using FlowJo.

The sorted cells were loaded to Chromium Next GEM Chip G (10x Genomics) on Chromium Controller (10x Genomics) for barcoding and cDNA synthesis. Amplification of the cDNA and the library construction was performed using Chromium Next GEM Single Cell 3' GEM, Library & Gel Bead Kit v3.1 or Chromium Next GEM Single Cell 3' Kit v3.1 (10x Genomics) for 3' profiling and Chromium Next GEM Single Cell 5' Kit v2 and Chromium Single Cell Human BCR Amplification Kit or Chromium Single Cell Human TCR Amplification Kit (10x Genomics) for 5' and VDJ profiling according to the manufacturer's protocol. The libraries were sequenced on NovaSeq6000 (Illumina).

**TCGA-THYM bulk RNA-seq analysis**. RNA-seq fastq files for thymoma were downloaded from the GDC Data Portal using gdc-client. Gene expression matrix quantified by HTSeq and clinical information was downloaded through an R package TCGAbiolinks. The detection of differentially expressed genes was performed by DESeq2[21] (1.30.1) with the design ~ primary_pathology_history_myasthenia_gravis after the removal of mean count below 5. For the visualization of a volcano plot, the lfcShrink function in DESeq2 was applied. Visualizations were performed by the plotPCA function in DESeq2, and R packages EnhancedVolcano, pheatmap, and ggplot2.

**WGCNA analysis**. A transformed matrix by the vst function in DESeq2 was used for WGCNA (1.71) analysis. The top 3000 genes in the variance of the vsd matrix were selected. Then, we calculated the adjacency using the adjacency function with power = 5, created Topological Overlap Matrix by TOMsimilarity, calculated the gene tree by hclust against 1 - TOM with method = "average", and conducted a dynamic tree cut with the following parameters; deepSplit = 2, pamRespectsDendro = FALSE, minClusterSize = 50. The eigengenes of each module were used for the correlation with clinical information. A pathway enrichment analysis was performed utilizing R packages clusterProfiler (3.16.1) and ReactomePA (1.32.0). Genes included in each module or included in the yellow module and with $log_2$ fold change > 1, and genes of each module were analyzed using the enrichPathway function.

**Immunoreceptors quantification**. The determination and quantification of TCR and BCR were performed by the MiXCR[63] (v3.0.3) analyze shotgun command with the options;–species hs–starting-material rna–only-productive. The Gini index for CDR3 amino acid sequences was calculated by an in-house program implemented in Python.

For HLA genotyping and quantification, we first aligned fastq reads on the hg38 reference genome using STAR (v2.7.2a). Then, HLA genotypes and expressions were extracted using arcasHLA[64] (v0.2.0) with IMGT.HLA database (3.24.0) with default parameters.

**Comprehensive virus detection from bulk RNA-seq**. The comprehensive viral quantification of RNA-seq was performed by a bioinformatics pipeline; VIRTUS[65] (v1.2.1), which was composed of fastp, STAR, and Salmon. First, we created indices using createindex.cwl with references downloaded from Gencode v33. Then, we quantified viruses using VIRTUS.PE.cwl with options–hit_cutoff 0–kz_threshold 0.3.

**Somatic mutation analysis of TCGA-THYM**. Mutation data of thymoma was downloaded using an R package, TCGAbiolinks (2.16.4). Visualization was performed by the oncoplot function implemented in an R package, maftools.

**Bioinformatics analysis of scRNAseq**. Sequenced reads were quantified by Cell Ranger (v5.0.0) with pre-built reference refdata-gex-GRCh38-2020-A downloaded at 10x GENOMICS' website. Quantified expressions were preprocessed and visualized using Scanpy[66] 1.7.2 and python 3.8.0. IGKV, IGLV, IGHV, IGLC,

TRAV, and TRBV genes were removed for the clustering and embedding for the removal of the effect of clonal expansion. Cells with mitochondrial genes were >20%, or detected genes <200 were filtered out, then preprocessed by sc.pp.normalize_per_cell with counts_per_cell_after=1e4, sc.pp.log1pp, retained highly variable genes, scaled using sc.tl.scale, and computed principal components using sc.tl.pca. The batch effect of samples was removed by the BBKNN[67] algorithm. Cells were embedded by UMAP using sc.tl.umap, clustered using sc.tl.leiden, and manually annotated. T cells, B cells, myeloid cells, and stromal cells were extracted, re-clustered from raw counts, and annotated manually through the same procedure where parameters were determined heuristically. The inference of the cell cycle was performed using the sc.tl.score_genes_cell_cycle function following the tutorial (https://nbviewer.jupyter.org/github/theislab/scanpy_usage/blob/master/180209_cell_cycle/cell_cycle.ipynb). The enrichment scores of gene sets such as GWAS-reported genes and WGCNA module genes were calculated by the sc.tl.score_genes function of Scanpy. A two-sided Mann–Whitney U-test was performed for scores of a cluster and that of others by the scipy.stats function. P-value correction for multiple tests was conducted using the statmodels package. Gene set enrichment analysis for clusters was performed by prerank test implemented in gseapy with scores calculated by sc.tl.rank_genes_groups with the option method=t-test_overestim_var.

To examine the difference in cell distribution in the thymus and blood, we used Bayesian estimation in consideration of the imbalance in the number of observed cells in the samples. For each cluster, we inferred the difference between $p_{thymus}$ and $p_{blood}$ using the following model;

$$x_{thymus} \sim Binomial\left(n_{thymus}, p_{thymus}\right) \tag{1}$$

$$x_{blood} \sim Binomial\left(n_{blood}, p_{blood}\right) \tag{2}$$

where,
x: number of detected cells for the cluster of an individual in the site.
p: probability that a cell is in the cluster of an individual in the site.
n: number of cells of an individual in the site.
The following was used as a prior distribution of p.

$$p_{thymus} \sim Uniform(0, 1) \tag{3}$$

$$p_{blood} \sim Uniform(0, 1) \tag{4}$$

The inference was conducted using a python package pymc3 (3.11.2) using 4 independent chains, 1,000 tuning iterations, and 25,000 additional iterations per chain. Trace plots and R_hat were used to assess the convergence.

For the extraction of thymus specific genes in CD4+ T cells (Fig. 4i), we examined the following cell types; CD4 $T_{NAIVE}$, Naive $T_{reg}$, Activated $T_{reg}$, CD4 $T_{CM}$ ($T_H0$), CD4 $T_{CM}$ ($T_H17$), CD4 $T_{CM}$ ($T_{FH}$), CD4 $T_{EM}$ ($T_H1$), CD4 $T_{EM}$ ($T_H1/17$), CD4 $T_{EMRA}$ ($T_H1$). Similarly, we used the following cell types in B cells (Supplementary Fig. 12c); Naive B cell, Plasmablast, Pre GC B cell, GC B cell, Memory B cell (I), Memory B cell (II), Thymic memory B cell, Unswitched memory B cell.

For the determination of RNA velocity, velocyto run10x was performed with the repeat file hg38_rmsk.gtf downloaded at the UCSC website. The projection of velocities was performed by scVelo (0.2.3)[68] following the same procedures and parameters as the official tutorial (https://scvelo.readthedocs.io/VelocityBasics/). For TCR analysis, we used the standard pipeline of scirpy (0.10.1)[69] according to the official tutorial (https://scverse.org/scirpy/latest/tutorials/tutorial_3k_tcr.html). Briefly, the distances of clonotypes were measured using the scirpy.pp.ir_dist function with parameters; metric="alignment", sequence="aa", cutoff=15 and defined clonotypes using scirpy.tl.define_clonotype_clusters with parameters; sequence="aa", metric="alignment", receptor_arms="all", dual_ir="any". Repertoire similarities were measured using the function scirpy.tl.repatoire_overlap. For the pathway enrichment analysis, a python package gseapy was used. Visualization was performed by functions of python packages; Scanpy, plotly (4.14.3), matplotlib (3.4.1), and seaborn (0.11.1). Details were described in codes deposited in the Github repository.

**Integration with public single-cell data**. H5ad files of scRNAseq data previously reported were downloaded respectively (PBMC: https://atlas.fredhutch.org/data/nygc/multimodal/pbmc_multimodal.h5seurat; normal thymus: 10.5281/zenodo.3711134). Data integration with public single-cell data was performed by the sc.tl.ingest function in Scanpy. For each reference dataset, we extracted highly variable genes, normalized and scaled the expression, and ingested our dataset to reference datasets similarly to our dataset. For the integration of TEC cells and B cells, we re-clustered cells from two datasets with the batch correction instead of the ingestion because some clusters were expected not to have their counterparts leading to the failure of the appropriate ingestion. We first concatenated our data with data from Park et al., removed the batch effect with BBKNN[67], and calculated correlations using sc.tl.dendrogram because TEC cells in thymoma were expected to be consist of a different set of cells from that in the normal thymus.

**Definition of tissue-restricted antigens (TRA)**. To define the list of tissue-restricted antigens (TRA), we used bulk RNA-seq data across tissues provided by

Genotype-Tissue Expression (GTEx) project (https://storage.googleapis.com/gtex_analysis_v8/rna_seq_data/GTEx_Analysis_2017-06-05_v8_RNASeQCv1.1.9_gene_tpm.gct.gz). We calculated the Gini index for mean TPM across tissues and extracted genes with Gini index > 0.8 and mean TPM at the maximum expressed tissue > 100 as TRAs (Supplementary Data 15).

**Inference of cell–cell interaction**. Cell–cell interaction was inferred by CellPhoneDB (2.1.7), which utilizes abundantly curated ligand-receptor pairs to measure the interactions within single-cell datasets. Statistical test was performed with the default parameters. Dot plots of ligand-receptor pairs were plotted by the cellphondb plot dot_plot function.

**Deconvolution of bulk RNA-seq**. A deep-learning-based deconvolution tool, Scaden[47] (v1.1.0) was used for the deconvolution of bulk RNA-seq datasets by TCGA. First, we created 30000 simulation datasets with scaden simulate by scaden simulate with option -n 30000. Second, count matrices of our single-cell dataset and TCGA thymoma dataset quantified by HTseq downloaded by TCGAbilinks were pre-processed by the scaden process command. Then, trained a network by the command scaden train with the option–steps 5000. Lastly, the bulk RNA-seq matrix was deconvoluted by scaden predict. Deconvoluted cell proportion was tested using a multiple linear regression provided as the formula.api.ols function by a python package statsmodels (0.12.0) with a model,
cells ~ MG + WHO + days_to_birth + Gender + 1.

**Curation of GWAS-reported genes**. We listed up GWAS-reported genes ($P < 5 \times 10^{-6}$) from previous reports[48–50]. Genes in HLA regions were excluded from the list. For eQTL and LD-aware gene mapping, LDexpress (https://ldlink.nci.nih.gov/?tab=ldexpress) in the LDlink suite[70] was used. We used all populations for the LD reference and all tissues from GTEx v8[71] for eQTL reference with the threshold, $R^2 \geq 0.1$, $P < 0.1$. For each reported locus, we selected a gene that possesses the smallest P-value.

**Statistical analysis**. All statistical analyses were performed in R (4.0.3) and python (3.8.0). FDR was obtained by the Benjamini–Hochberg procedure implemented by a python package statsmodels (0.12.0). Pearson's correlation used for Fig. 6f was calculated using a python package pingouin (0.3.8). The visualization of a network was performed using Cytoscape (3.8.0)[72]. All other statistical analyses are detailed in the respective sections of the article.

**Reporting summary**. Further information on research design is available in the Nature Research Reporting Summary linked to this article.

## Data availability

Single-cell data can be explored interactively and downloaded in SingleCellPortal (https://singlecell.broadinstitute.org/single_cell/study/SCP1532). The raw sequence data for single-cell RNA-seq analysis was deposited in JGA ("JGAS000482[https://ddbj.nig.ac.jp/resource/jga-study/JGAS000482]"). TCGA data is available on dbGaP accession "phs000178[https://www.ncbi.nlm.nih.gov/projects/gap/cgi-bin/study.cgi?study_id=phs000178.v11.p8]". All other data are provided in the article and its Supplementary files or from the corresponding author upon reasonable request. Source data are provided with this paper.

## Code availability

All source codes were deposited in the GitHub repository (https://github.com/yyoshiaki/MG_thymoma_Manuscript_2021).

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

## Acknowledgements

We thank Y. Tachibana, K. Funakoshi, and A. Harada for supporting the annotation of single-cell data. We thank T. Sawamura, M. Nihei at the Department of Pathology, Osaka University for supporting the preparation of slides for histological assessments. We thank M. Okumura for his critical advice on the study. This study was supported by the Center for Medical Research and Education, Graduate School of Medicine, Osaka University. We acknowledge the NGS core facility of the Genome Information Research Center at the Research Institute for Microbial Diseases of Osaka University for the support in RNA sequencing. Some illustrations were generated with BioRender.com. The results published here are in part based upon data generated by the TCGA Research Network: https://www.cancer.gov/tcga. This work was supported by Grants-in-Aid by Japanese Society for the Promotion of Science (JSPS) for Specially Promoted Research 16H06295 to S.S., by the Core Research for Evolutional Science and Technology (CREST, no. 17 gm0410016h0006) program from the Japan Science and Technology Agency to S.S. and by Leading Advanced Projects for medical innovation (LEAP, no. 18 gm0010005h0001) from Japan's Agency for Medical Research and Development (AMED) to S.S.

## Author contributions

Y.Y., T.O., N.O., and H.Mo. designed all experiments; Y.Y., H.Mu., K.K., M.Ko, Y.N., and M.A performed experiments under the supervision of T.O., M.Ki., S.N. and N.O; Y.Y., E.T., S.Su., and Y.T. performed bioinformatics analysis; Y.Y. and K.K. diagnosed thymoma pathology under the supervision of S.N. and E.M.; Y.S. and S.F. collected samples for analysis; D.M. and D.O. performed library construction and sequencing; Y.Y., E.T. and H.Mu. prepared the figures; Y.Y. and N.O. drafted the manuscript; T.O., H.Mo., E.M., N.O. and S.Sa. supervised the study; S.T. and M.O. provided expert guidance on the manuscript; All authors critically reviewed and edited the final version of the manuscript.

## Competing interests

The authors declare no competing interests.
