## [Peer Review File · Nature Communications]

Myasthenia gravis-specific aberrant neuromuscular gene expression by medullary thymic epithelial cells in thymomaREVIEWER COMMENTS

Reviewer #1 Myasthenia Gravis (Remarks to the Author):

This is a study to investigate the pathogenesis of thymoma-associated myasthenia gravis. The authors have provided comprehensive perspectives on autoimmunity associated with thymoma, using the state-of-the-art scientific methods. I believe this study could enlarge the point of view about neurological autoimmune disease. I only have a few questions about this study.

1. Why did you include a patient, who had received immunosuppressive therapy preoperatively, in single-cell RNA sequencing experiments? Did you have any concern about inaccuracy of data due to immunosuppressant usage? Please indicate the dose of the immunosuppressant and the duration of the therapy.

2. Since thymic epithelial cells, especially nmTECs, play a crucial role in autoantibody production, it seems that type B3 thymoma, which holds more epithelial cells than type B2 thymoma, should have a stronger correlation with MG. However, type B2 thymoma is more associated with MG than type B3, in fact. Could you explain why?

3. Even though thymoma-associated MG is strongly associated with autoantibodies targeting acetylcholine receptor, authors found only a slight increase of CHRNA1 expression, compared to other genes such as RYR3 and GABRA5. It seems that higher expression of CHRNA1 could lead to more chances for nmTECs to present self-antigens and produce autoantibodies. Could you explain why MG-associated thymoma shows only a slight increase of CHRNA1 expression?

Reviewer #2 Thymoma (Remarks to the Author):

General comments:

In their manuscript, Dr Yasumizu and collaborators analyzed cell profiles from thymoma associated with Myasthenia Gravis (MG) or not. First, they used bulk RNAseq data from thymoma samples enrolled by the Cancer Genome Atlas (TCGA) to define gene expression profiles. They compared thymoma with and without MG and identified differentially expressed genes and also "gene modules" associated with MG. They described the up-regulation of neuromuscular-related molecules linked to MG.

Next, they performed SC-RNAseq on thymoma and PBMC derived from four MG patients. All data from thymic tissue and PBMC were pooled to process unsupervised analyses. They identified, in particular, a thymic epithelial cell cluster that also expressed neuromuscular-related molecules. These thymic epithelial cells were named nmTECs and the authors suggested that nmTECs play a significant role in MG pathogenesis via ectopic expression of neuromuscular molecules.

In addition, they analyzed myeloid cells, B cells, and T cells in MG-thymoma. Thanks to the analysis of cell-cell interaction, MG pathology across cell types and validation by IHC, they suggested that MG-thymoma exhibited microenvironments dedicated to autoantibody production, including ectopic germinal center formation, T follicular helper cell accumulation, and type 2 conventional dendritic cell migration.

Bulk RNAseq and SC-RNAseq data were processed by multiple approaches that are for some of them a bit out of my area of expertise. However, I have some concerns that they were able to analyze stromal cells other than TECs because they applied a cell sorting strategy for TEC cells. In addition, the number of cells analyzed for each subtype should be given, in particular for TEC. These analyses were conducted under the supervision of researchers with good expertise in Myasthenia Gravis and Thymoma.

Below are listed comments to further improve the manuscript.

Introduction:

In the introduction, a sentence should be modified "In addition, dysregulation of the thymus,

including thymoma and thymic hyperplasia, are frequently associated not only with MG but also neurological disorders, including encephalitis, which are caused by a wide range of autoantibodies” This is true for thymoma but not that much for MG patients without thymoma.

Methods and Results:

Were thymic tissues involved in SC-RNAseq analyses thymoma biopsies or adjacent tissues? This question is about the fact that the presence of germinal centers is so far considered to be in the adjacent tissue of thymoma.

In table S7: the MG score should be indicated as one patient has an anti-AChR titer below 0.3 and could be considered as not MG.

The cell sorting strategy was developed to collect thymic epithelial cells (CD45^{low} EpCAM⁺) or immune cells (non-adherent cells). How to explain the presence of stromal cells such as endothelial cells and fibroblasts in the final analyses. Similarly, how to explain the detection of macrophages and DCs. Was it possible to recover some of these cells just in the pool of thymic cells obtained from the mechanical disruption?

For SC-RNAseq analyses, they profiled 33,839 cells from thymoma and 30,810 cells from PBMCs. Could the detail per patient be given for example table 7 for thymic cells and PBMCs.

Figure 3a: The number of stromal cells and consequently of TECs seems very low. The total number of TECs that were analyzed for cTEC, mTEC1, mTEC2, and nmTEC should be given. The clusters defining the different TEC subpopulations were not well delineated how do they explain that?

Figure 4e: Authors should specify the number of B cells that were analyzed by SC-RNAseq in the thymus

Figure 4j: the calculation of the proportion is not clear

Figure 5e: CD31 is also expressed on thymocytes and the staining did not seem to correspond to endothelial blood vessels.

Line 257: the figure mentioned should be Fig. 4e?

Line 301: The concerned figure should be indicated.

Lines 327-328, the authors suggest that “These observations suggest that nmTECs may promote angiogenesis via the interaction of vascular endothelial cells”. Could these endothelial cells correspond to high endothelial venules as observed by Lefevre et al?

Lines 335-337: It is not clear if the authors were speaking of mTEC (I) ou (II) as there was no graph for mTEC.

Lines 341-342: the authors suggest that “The most significantly associated cell population to MG was nmTECs, followed by GC B cells and cDC2s (Fig. 6a-c)”. The interpretation of this figure is not clear-cut as no significant values were given.

Line 1024: the authors mentioned a GWAS for EOMG and LOMG. The GWAS should be explained and the EOMG and LOMG form of the disease explained.

Extended Data Fig. 9a: the figure legend should be more detailed

Discussion:

The axis CXCL12-CXCR4 is mentioned on several occasions, even in the abstract. This could be linked to the publication of Weiss et al. in immunobiology 2013 that investigate this axis in MG

patients.

Thymoma-associated MG is known to be associated with an interferon type I (IFN-I) signature and also the presence of autoantibodies against IFN- α . Did the authors search for this signature in the dysregulated genes or pathways?

Reviewer #3 systems immunology / single cell sequencing (Remarks to the Author):

The authors harnessed the power of high-throughput sequencing methods in singular and combination, to investigate the microenvironment of co-existed MG and thymoma. The authors conducted thorough data analysis stemmed from well-planned experiments. I appreciate that the methodology on data analysis and cell separations were well documented. For the clarification of the manuscript and soundness of the study, I have the following suggestion.

1. In Figure 1A, there seems to be a considerable overlapping between MG and non-MG profiles according to your definition. Alternatively speaking, profile of MG and non-MG are not separable. How could you explain this observation? I suggest to include analysis of samples from health donor here, in the beginning of this manuscript, to present the differences of HD and thymoma patients.
2. Figure g, h. In this part of the analysis, the authors selected several hits from the selective pathways. Are these pathways ranking the top in pathway analysis of this dataset? And is that the reason that they were selected? If so, please provide supporting data, such as pathway analysis and ranking together with other pathways that were not selected.
3. As author stated in line 164 "The diversity of TCR was mostly unchanged between them", I don't agree that it's proper to draw the conclusion in line 181-182.
4. I was puzzled by the justification and benefits of analyzing PBMC and thymocytes by mixing them together in figure 2. I understand that the authors would like to examine a broad range of immune cell types. However, the mixing of the two samples are not necessary in my opinion.
5. Figure 3. Could the authors specify the cell numbers in each clusters of TEC? It seems to me that only very few number of cells in the newly defined cluster. Could the low cell number be the reason for low p value in the analysis and lead to the identification of the new cluster?
6. I'm curious whether the authors observed any of T cell exhaustion markers that was distinguishable in nmTEC cluster.
7. Please provide a complete list of genes used for delimitation of the all cell types in the single cell analysis.
8. The authors provided a plethora of analysis and information in searching for the unique markers of targeted disease. However, it's not clear to me how all these information were narrowed down to GABRA5 and KRT6.
9. Typo in Figure 6, "INF-gamma" should be IFN gamma.

We thank all reviewers for their favorable view to the manuscript, and also for many helpful and valuable comments. We have carefully addressed each comment by in some cases re-analyzing the data, and in others adding descriptions, figures, and tables to make the statement clear. We also have made the following appropriate improvements in response to the editor's indications through the response to the reviewers. Thanks to the reviewers' comments, we now believe that our revised manuscript has been fully improved. Throughout the responses, our responses are highlighted in blue.

Reviewer #1 Myasthenia Gravis (Remarks to the Author):

This is a study to investigate the pathogenesis of thymoma-associated myasthenia gravis. The authors have provided comprehensive perspectives on autoimmunity associated with thymoma, using the state-of-the-art scientific methods. I believe this study could enlarge the point of view about neurological autoimmune disease. I only have a few questions about this study.

We greatly appreciate the reviewer's positive comments.

1. Why did you include a patient, who had received immunosuppressive therapy preoperatively, in single-cell RNA sequencing experiments? Did you have any concern about inaccuracy of data due to immunosuppressant usage? Please indicate the dose of the immunosuppressant and the duration of the therapy.

Thank you for the critical comment. MG23 patient received methylprednisolone pulse therapy (250mg/day for 3 days; 500mg/day for 3 days) a few months before sample collection, followed by oral prednisone (5mg/day). We considered that the cell profiles of stromal cells in the thymus were relatively maintained in the short period of immunosuppressant administration, and thus enrolled the patient only for the profiling of thymic stromal cells. In addition, the purpose of the single-cell analysis was the identification of cells and marker genes for MG pathogenesis rather than the assessment of numbers in each cell type. Under this situation, we believe that the immunosuppressant usage did not skew the results. We described the detailed clinical information for each patient in **Supplementary Table 7**.

2. Since thymic epithelial cells, especially nmTECs, play a crucial role in autoantibody production, it seems that type B3 thymoma, which holds more epithelial cells than type B2 thymoma, should have a stronger correlation with MG. However, type B2 thymoma is more associated with MG than type B3, in fact. Could you explain why?

Thank you for raising an important point. Indeed, if only nmTEC contributed to the onset of the disease, as the referee pointed out, the complication rate would be higher in type B3 and C, which have more epithelium, compared to types B1 and B2. On the other hand, a certain number of MG complications have been observed in B2 and B1 types in reality, suggesting that even a small proportion of nmTEC

can contribute to MG pathogenesis. In addition, the presence of lymphocytes may be necessary for the development of MG, because no complications are seen in type C. To clarify this point, the manuscript has been revised as follows; “Regarding WHO classification, MG complications were observed not only in B3, which has a high epithelium, but also in B2 and B1 (Extended Data Fig. 2a). ... These results suggest that the accumulation of neuromuscular-related antigens induces a pre-disease state and is not a sufficient condition for MG pathogenesis.” in **Discussion** (Line 455-461), “MG was associated with multiple types except for type C (Extended Data Fig. 2a), with its peak at type B3 (MG complication rate: 54.5%) and B2 (53.8%), whereas a previous report observed the peak at type B2 (71.1%).” in **Results** (line 99-101), and “MG complication rates were 21.4% in type A, 23.5% in type AB, 0% in type A;AB, 14.3% in type B1, 0% in type B1;B2, 53.8% in type B2, 0% in type B2;B3, 54.5% in type B3, and 0% in type C.” in **Extended figure legends** (Extended Data Fig. 2a).

3. *Even though thymoma-associated MG is strongly associated with autoantibodies targeting acetylcholine receptor, authors found only a slight increase of CHRNA1 expression, compared to other genes such as RYR3 and GABRA5. It seems that higher expression of CHRNA1 could lead to more chances for nmTECs to present self-antigens and produce autoantibodies. Could you explain why MG-associated thymoma shows only a slight increase of CHRNA1 expression?*

We totally agree with the referee's comments, and we were also surprised at the moderate expression change of *CHRNA1* compared with those of *RYR3* and *GABRA5*. Based on the results, we are now inferring that the conspicuous symptoms caused by the disruption of AChR function may be derived from a combination of the amount of AChR as the antigen, the physiological accessibility of autoantibody to the antigen, and the biological importance of AChR. To reflect the reviewer's comment, we added the descriptions as; "In addition, while the most frequent complication of thymoma is anti-acetylcholine receptor antibody-positive myasthenia gravis, the increase of the *CHRNA1* expression was milder than other neuromuscular antigens such as *GABRA5* and *RYR3*. The moderate increase of *CHRNA1* expression may be sufficient for causing serious symptoms due to the accessibility of acetylcholine receptor antibodies⁵⁴ and the importance of their biological functions⁵⁵." in **Discussion** (line 422-427).

Reviewer #2 Thymoma (Remarks to the Author):

General comments:

In their manuscript, Dr Yasumizu and collaborators analyzed cell profiles from thymoma associated with Myasthenia Gravis (MG) or not. First, they used bulk RNAseq data from thymoma samples enrolled by the Cancer Genome Atlas (TCGA) to define gene expression profiles. They compared thymoma with and without MG and identified differentially expressed genes and also “gene modules” associated with MG. They described the up-regulation of neuromuscular-related molecules linked to

MG.

Next, they performed SC-RNAseq on thymoma and PBMC derived from four MG patients. All data from thymic tissue and PBMC were pooled to process unsupervised analyses. They identified, in particular, a thymic epithelial cell cluster that also expressed neuromuscular-related molecules. These thymic epithelial cells were named nmTECs and the authors suggested that nmTECs play a significant role in MG pathogenesis via ectopic expression of neuromuscular molecules.

In addition, they analyzed myeloid cells, B cells, and T cells in MG-thymoma. Thanks to the analysis of cell-cell interaction, MG pathology across cell types and validation by IHC, they suggested that MG-thymoma exhibited microenvironments dedicated to autoantibody production, including ectopic germinal center formation, T follicular helper cell accumulation, and type 2 conventional dendritic cell migration.

Bulk RNAseq and SC-RNAseq data were processed by multiple approaches that are for some of them a bit out of my area of expertise. However, I have some concerns that they were able to analyze stromal cells other than TECs because they applied a cell sorting strategy for TEC cells. In addition, the number of cells analyzed for each subtype should be given, in particular for TEC.

These analyses were conducted under the supervision of researchers with good expertise in Myasthenia Gravis and Thymoma.

Thank you so much for the thoughtful comments and the kind suggestions.

Below are listed comments to further improve the manuscript.

Introduction:

In the introduction, a sentence should be modified “In addition, dysregulation of the thymus, including thymoma and thymic hyperplasia, are frequently associated not only with MG but also neurological disorders, including encephalitis, which are caused by a wide range of autoantibodies” This is true for thymoma but not that much for MG patients without thymoma.

Thank you for the comments. We modified the sentence as follows; “In addition, dysregulation of the thymus, especially thymoma, is frequently associated not only with MG but also neurological disorders, including encephalitis, which is caused by a wide range of autoantibodies.”

Methods and Results:

Were thymic tissues involved in SC-RNAseq analyses thymoma biopsies or adjacent tissues? This question is about the fact that the presence of germinal centers is so far considered to be in the adjacent tissue of thymoma.

We agree with the reviewer’s viewpoint that the sampling can bias the germinal center (GC) existence. In this study, all specimens were derived from surgically removed thymoma, which included a small marginal area of them. Consistent with the reviewer's comment, it is possible that the GCs tended to

be derived from the specimen margins. We noticed that GCs tended to exist in the peripheral region of the thymoma in the microscopic observation of 63 thymoma specimens, although it was not objectively demonstrated. To clarify this point, we have rephrased the sentence as; “The sections were mainly consisted of thymoma with a small proportion of marginal regions.” in **Methods** (line 522-523).

In table S7: the MG score should be indicated as one patient has an anti-AChR titer below 0.3 and could be considered as not MG.

Thank you for the comment and for raising an important point. As Reviewer suggested, we added the QMG score to Supplementary Table 7. Regarding the anti-AChR titer, the standard value was 0.2 nmol/L or less, and 0.3 nmol/L was treated as positive in this study. In addition, we enrolled anti-AChR positive patients for single-cell analysis regardless of QMG score. We have clarified this point in the manuscript by describing thymoma from anti-AChR antibody-positive patients as “MG-type” thymoma.

The cell sorting strategy was developed to collect thymic epithelial cells (CD45^{low} EpCAM⁺) or immune cells (non-adherent cells). How to explain the presence of stromal cells such as endothelial cells and fibroblasts in the final analyses. Similarly, how to explain the detection of macrophages and DCs. Was it possible to recover some of these cells just in the pool of thymic cells obtained from the mechanical disruption?

We thank Reviewer for pointing out the important point. The myeloid lineage was included in the immune cell pools. We used the lower threshold of EPCAM staining for TEC sorting due to the low number of EPCAM positive cells, therefore fibroblasts and endothelial cells were included in the single-cell pool. Also, these cells were classified correctly based on the marker gene expression, as shown in Extended Data Figure 5. We attempted to provide a reasonable explanation to readers by adding the following sentence in Methods; “To increase the collection of EPCAM⁺ cells, we used the lower threshold of EPCAM for the gating, resulting in the inclusion of fibroblasts and endothelial cells.” (line 535-537).

For SC-RNAseq analyses, they profiled 33,839 cells from thymoma and 30,810 cells from PBMCs. Could the detail per patient be given for example table 7 for thymic cells and PBMCs.

Thank you for the suggestion. We added the number of thymic cells and PBMCs profiled in the scRNAseq analysis to **Supplementary Table 7**, in addition to Extended Data Fig. 5a.

Figure 3a: The number of stromal cells and consequently of TECs seems very low. The total number of TECs that were analyzed for cTEC, mTEC1, mTEC2, and nmTEC should be given.

We agree that we should provide information about the number of cells for each cluster. The numbers of cells for TEC clusters were cTEC:112, mTEC (I):180, mTEC (II):106, nmTEC:24. We have provided the number of cells for each cluster in **New Supplementary Table 8**.

The clusters defining the different TEC subpopulations were not well delineated how do they explain that?

Thank you for the comments. We have described marker genes for TEC subpopulations in Extended Data Fig.6a. In addition, to provide detailed information on gene profiles, we performed gene set enrichment analysis using REACTOME gene sets and added **New Extended Data Figure 6h**, and **New Supplementary Data Table 12-14**.

New Extended Data Figure 6h

Volcano plot showing REACTOME gene sets enriched in TEC clusters (Supplementary Table 12-14).

Figure 4e: Authors should specify the number of B cells that were analyzed by SC-RNAseq in the thymus

Thank you for the comment. Regarding B cells, 1061 and 8888 cells were derived from the thymus and the periphery, respectively. We have provided the number of cells for all clusters from the thymus and the periphery in **New Supplementary Table 8**.

Figure 4j: the calculation of the proportion is not clear

For the calculation of the proportion, we are using the Bayesian estimation framework to consider the imbalance in the number of observed cells depending on the samples. The description of the procedure has been provided in Methods. In addition, we modified the Figure legend as; “Cell proportion of each T cell cluster in thymoma and peripheral blood. Error bars show 98% highest density interval. *FDR < 0.05. (Methods)”.

Figure 5e: CD31 is also expressed on thymocytes and the staining did not seem to correspond to endothelial blood vessels.

Thank you for the suggestion. We changed the representative image in Figure 5e.

(New Figure 5e)

Line 257: the figure mentioned should be Fig. 4e?

Thank you for your suggestion. Since we intended to show the existence of GC B-cells as a single-cell cluster, we cited Fig.4a at this location. The difference of cell number (Fig. 4e) was mentioned in the latter section (page 8, line 273).

Line 301: The concerned figure should be indicated.

Thank you, we moved the indication to the appropriate position (line 312).

Lines 327-328, the authors suggest that “These observations suggest that nmTECs may promote angiogenesis via the interaction of vascular endothelial cells”. Could these endothelial cells correspond to high endothelial venules as observed by Lefevre et al?

Thank you for the suggestion. This report empowers our observations. We have cited the article in the manuscript as follows; “These observations suggest that nmTECs may promote angiogenesis via the interaction of vascular endothelial cells, which is consistent with the previous reports¹⁷.”

Lines 335-337: It is not clear if the authors were speaking of mTEC (I) or (II) as there was no graph for mTEC.

We agree with the reviewer that mTEC(I) and (II) should be clearly stated. We have described this point by discriminating mTEC (I) and (II) as follows; “Among cell populations, cTECs were accumulated in WHO type A; mTEC(I) in type A, B3, C; mTEC(II) in type A; and immature T cells in type B1 thymoma (Extended Data Fig. 9a).” in **Results** (line 348).

Lines 341-342: the authors suggest that “The most significantly associated cell population to MG was nmTECs, followed by GC B cells and cDC2s (Fig. 6a-c)”. The interpretation of this figure is not clear-cut as no significant values were given.

Thank you for the comment. We rephrased the Figure legend as;

“Volcano plot showing the association with MG for deconvoluted cell proportions, which were calculated from TCGA bulk RNA-seq dataset with the reference defined in our scRNAseq analysis. Coefficients and p-values were calculated with multiple regression (Methods). Red dot FDR < 0.05, orange dots FDR < 0.2.”. Together with the description in Methods (line 623-638), we now believe that the association analysis in Figure 6a has been described enough.

Line 1024: the authors mentioned a GWAS for EOMG and LOMG. The GWAS should be explained and the EOMG and LOMG form of the disease explained.

Thank you. The suggested correction has been made.

Extended Data Fig. 9a: the figure legend should be more detailed

We added the detailed description in the legend; “Cell deconvolution analysis was performed for TCGA thymoma bulk RNA-seq samples with single-cell annotations as the reference (a,b). Scaden⁴⁷ was used for the deconvolution.”

Discussion:

The axis CXCL12-CXCR4 is mentioned on several occasions, even in the abstract. This could be linked to the publication of Weiss et al. in immunobiology 2013 that investigate this axis in MG patients.

Thank you for the suggestion. We have cited this article in **Discussion**.

Thymoma-associated MG is known to be associated with an interferon type I (IFN-I) signature and also the presence of autoantibodies against IFN- α . Did the authors search for this signature in the dysregulated genes or pathways?

Thank you so much for raising the important point. We succeeded in showing the significant association with the IFN α signaling pathway in nmTEC as well as the IFN γ signaling. Although we could not assert the biological mechanism of the production of anti-IFN α antibodies, it may be possible that the overexpression of IFN α by nmTEC induces the autoantibody similarly to other neuromuscular molecules. We have added "Regulation of IFNA signaling" in **Figure 3i** and **Figure3j**, and its descriptions in the manuscripts.

new Figure 3i

The label for Regulation of IFNA signaling was added.

new Figure 3j

Violin plot for Regulation of IFNA signaling was added.

Reviewer #3 systems immunology / single cell sequencing (Remarks to the Author):

The authors harnessed the power of high-throughput sequencing methods in singular and combination, to investigate the microenvironment of co-existed MG and thymoma. The authors conducted thorough data analysis stemmed from well-planned experiments. I appreciate that the methodology on data analysis and cell separations were well documented. For the clarification of the manuscript and soundness of the study, I have the following suggestion.

I greatly appreciate the reviewer's comments and suggestions.

1. In Figure 1A, there seems to be a considerable overlapping between MG and non-MG profiles according to your definition. Alternatively speaking, profile of MG and non-MG are not separable. How could you explain this observation? I suggest to include analysis of samples from health donor here, in the beginning of this manuscript, to present the differences of HD and thymoma patients.

Thank you for the critical comments. We respond to the comment by splitting it into two parts.

Part 1: Adding normal thymus samples

We added normal thymus samples collected from the child and fetal donors. Unexpectedly, the transcriptome profiles of normal thymus and thymoma were discriminative, and PC1 and PC2 were greatly affected by the difference between tumor and normal rather than tumor profiles (**Figure A for reviewer #3**). PC1 represented a high number of epithelial cells and PC2 a high number of lymphocytes. Type C is the most distant from normal, and A, AB, B1, B2, and B3 are equally distant from normal. Since this is not an article about tumorigenesis in thymoma, we decided not to discuss it in detail. We would be glad if the reviewer kindly accepted our standpoint.

Figure A for Reviewer #3.

PCA plots for transcription profile of thymomas from 116 patients and normal thymus from 7 donors. The left panel shows the disease status, MG or non-MG, and the right panel shows WHO classification based on histology or normal. Normal samples are indicated by a pink circle.

[Methods for Figure A: Sequence data (GSE18927 and GSE16256) from the Roadmap project and UCSD Human Reference Epigenome Mapping Project were used as normal thymus samples. Followed by the procedures for RNAseq in TCGA consortium (https://docs.gdc.cancer.gov/Data/Bioinformatics_Pipelines/Expression_mRNA_Pipeline), download

sequences were aligned on the reference genome provided by Gencode v22 using STAR-2.4.2a with two-pass mode, and genes were quantified using HTSeq-0.6.1p1. Gene expression matrices were combined with the TCGA thymoma dataset and plotted PCA using DESeq2 as described in the Methods.]

Part 2: MG-nonMG separation

The data of TCGA is only labeled MG and non-MG, and if there is information of anti-AChR Abs, for example, as in Fig. 6e-j, sensitivity of the MG detection would be improved, and pre-symptomatic cases of MG might be detected. Also, as mentioned in the discussion (Line 453-472), we believe that multiple factors are necessary for the development of MG. Although MG-type thymoma appeared as a group in the PCA plot, only a certain number of them developed MG at the point of surgery. To describe this point clearer, we have modified **Discussion** (line 454-472) as follows;

“Our data showed that some patients with high expression of neuromuscular genes did not develop MG (Fig. 1a,c)... Histological analysis also showed that GABRA5-positive, or nmTEC marker-positive, cells were present in some acetylcholine receptor antibody-negative patients (Fig. 6e-j). These results suggest that the accumulation of neuromuscular-related antigens induces a pre-disease state and is not a sufficient condition for MG pathogenesis. ... Further analysis will be required for addressing the stepwise development of MG.”

2. Figure g, h. In this part of the analysis, the authors selected several hits from the selective pathways. Are these pathways ranking the top in pathway analysis of this dataset? And is that the reason that they were selected? If so, please provide supporting data, such as pathway analysis and ranking together with other pathways that were not selected.

Thank you very much for your comments. We inferred that the reviewer's comment was about Figure 3i. We selected several significant pathways that should be discussed and addressed in the text and Fig 3j-k. We have added a **new Supplementary Table 11** for the statistics of Fig 3i.

3. As author stated in line 164 “The diversity of TCR was mostly unchanged between them”, I don't agree that it's proper to draw the conclusion in line 181-182.

Thank you for the comment. To measure the diversity of immune receptors, we calculated the Gini index of CDR3 usage as shown in Extended Data Figure 3a. Two-sided Mann Whitney's U-test indicated that the Gini indexes of TRA, TRB, TRD and TRG were not different between MG and non-MG thymoma samples whereas IGH, IGL and IGK were. Therefore, we concluded that the diversity of TCR was unchanged between them. To avoid confusion, we rephrased the sentence as follows; “The diversity of TCR represented by the Gini index was mostly unchanged between them (Extended Data Fig. 3a), but ...” in **Results** (line 161-163).

4. I was puzzled by the justification and benefits of analyzing PBMC and thymocytes by mixing them together in figure 2. I understand that the authors would like to examine a broad range of immune cell types. However, the mixing of the two samples are not necessary in my opinion.

Thank you for pointing out the important point. Indeed, as the reviewer mentioned we also considered analyzing the datasets derived from thymoma and PBMCs separately. As immune cells are circulating between the thymus and the peripheral blood, we would like to know the trafficking by performing a common clustering rather than separate identification. As a result, we could see the traffic of GCs, myeloid cells as shown in Figure 4 and Extended Data Figure 7 and succeed in capturing thymus-specific genes in T cells by stratifying by clusters as in Figure 4k and Extended Data Figure 8. We also detected thymus-specific clusters such as CD8+ Trm in the analysis, therefore we considered that the strategy fitted well with this study. For the clarification of this point, we have described it as follows; “To analyze the similarity and differences between thymic cells and PBMCs, clustering was performed for the pooled cells.” in **Results** (line 193-194).

5. Figure 3. Could the authors specify the cell numbers in each clusters of TEC? It seems to me that only very few number of cells in the newly defined cluster. Could the low cell number be the reason for low p value in the analysis and lead to the identification of the new cluster?

Thank you for the suggestion. We have added the number of cells for each cluster in the new **Supplementary Table 8**. As the reviewer suggested, the single-cell analysis alone may not have been so powerful to identify new cell types. However, we believe that the integration with large-scale bulk RNAseq and histological observations provided enough evidences for the identification of the new cluster.

6. I'm curious whether the authors observed any of T cell exhaustion markers that was distinguishable in nmTEC cluster.

Thank you for the comments. We investigated exhaustion-related molecules on T-cells and the ligands on stomal cells (**Figure B for Reviewer**). Several cells classified as CD8 Trm and CD8 Tem expressed exhaustion-related molecules and cTECs expressed CD274 (PDL1) molecules. They suggest that anti-tumor immunity and exhaustion by TCR-stimulation occurs in MG-type thymoma. Because we made a full effort to share single-cell data as the resources (raw sequence data in NBDC and processed data in Single Cell Portal), we hope researchers interested in T-cell exhaustion and another phenomenon in the thymoma can freely get insight from the resources.

Figure B for Reviewer #3

Dot plot of gene expression of T-cell exhaustion-related molecules on T-cells and the ligands on stromal cells.

7. Please provide a complete list of genes used for delimitation of the all cell types in the single cell analysis.

Thank you for the comment. We agree that the expression of curated markers for each cell type would be informative in addition to the differentially expressed genes for each cluster. We have added a marker gene list calculated by t-test with overestimated variance in New Supplementary Table 10. We note that the calculations were performed for the very global clusters, and therefore the matrix contains redundancies and ambiguities.

8. The authors provided a plethora of analysis and information in searching for the unique markers of targeted disease. However, it's not clear to me how all these information were narrowed down to GABRA5 and KRT6.

Thank you for raising an important point. In practice, we first searched MG-thymoma-associated genes defined as the yellow module genes in Fig.1b. Among these genes, we selected genes log2 fold change between MG and non-MG thymoma in TCGA bulk RNAseq dataset was above 1 (212 genes remained). Then, we selected genes with mean expression in the nmTEC cluster above 0.3 in our single-cell dataset. Several genes such as NEFM, KRT6 and GABRA5 possessed a significant increase in nmTEC

compared with other TEC cells with stable expression (**New Extended Data Figure 6b**). Lastly, KRT6 and GABRA5 were selected because of the availability of good antibodies. To clarify this point, we added the sentence “Among the yellow module genes, we selected KRT6 and GABRA5 as marker genes of nmTECs in the following criteria; 1) the expression was increased in MG patients in TCGA bulk RNAseq dataset, 2) in the scRNAseq dataset, stably and preferentially expressed in nmTECs; and 3) availability of commercial antibodies (Extended Data Fig. 6b.” in **Results** (line 213-217) with **New Extended Data Figure 6b**.

New Extended Data Figure 6b

Bar plot for the yellow module genes with log₂ fold change (MG vs. nonMG in TCGA bulk RNAseq dataset) > 1 and mean expression nmTEC > 0.3 in nmTECs. log₂(fold change) was calculated by comparison with other TEC clusters. Mean expression in nmTEC was represented by color.

9. Typo in Figure 6, “*INF-gamma*” should be *IFN gamma*.

Thank you. We corrected the typo.

REVIEWERS' COMMENTS

Reviewer #1 (Remarks to the Author):

The authors have answered the comments precisely and revised the manuscript properly. I have no other comments and I believe the manuscript is entitled to the journal.

Reviewer #2 (Remarks to the Author):

In the revised version of their manuscript, Dr. Yasumizu and collaborators have answered all my questions and improved the manuscript accordingly. The manuscript seems fine for publication based on my "limited knowledge" of bioinformatics analyses.

As with many SC-RNAseq analyses, this manuscript is primarily descriptive but the information is important to the research community.

Reviewer #3 (Remarks to the Author):

In this manuscript, the authors identified ectopic expression of a set of neuromuscular molecules in MG thymoma. The authors first profiled thymoma specific genes using bulk RNA-seq of more than 100 thymoma samples. Subsequently, they conducted scRNAseq of thymoma samples and PBMCs, and identified unique characteristics of the cells, including B cell maturation and T cell polarization. Then they performed immunohistological test of MG thymomas to verify the biomarkers like GABRA5 and KRT6, and defined a new subset of TEC.

I appreciate that the authors supplemented with the cell numbers in each clusters TEC used for scseq analysis, since the extreme low number of the cells wouldn't provide enough power. Also, I appreciate the explanation for the specific experimental process used in validation of genes from sequencing technologies, ie. Mixing PBMC and thymocytes. I think that the authors addressed all my concerns and clarified the issues that caused my confusion.

Reviewer #1 (Remarks to the Author):

The authors have answered the comments precisely and revised the manuscript properly. I have no other comments and I believe the manuscript is entitled to the journal.

Thank you very much for the careful review and favorable comments.

Reviewer #2 (Remarks to the Author):

In the revised version of their manuscript, Dr. Yasumizu and collaborators have answered all my questions and improved the manuscript accordingly. The manuscript seems fine for publication based on my "limited knowledge" of bioinformatics analyses.

As with many SC-RNAseq analyses, this manuscript is primarily descriptive but the information is important to the research community.

Thank you so much for the comments in the perspective of a specialist of thymoma. We believe the keen comments greatly improved manuscripts.

Reviewer #3 (Remarks to the Author):

In this manuscript, the authors identified ectopic expression of a set of neuromuscular molecules in MG thymoma. The authors first profiled thymoma specific genes using bulk RNA-seq of more than 100 thymoma samples. Subsequently, they conducted scRNAseq of thymoma samples and PBMCs, and identified unique characteristics of the cells, including B cell maturation and T cell polarization. Then they performed immunohistological test of MG thymomas to verify the biomarkers like GABRA5 and KRT6, and defined a new subset of TEC.

I appreciate that the authors supplemented with the cell numbers in each clusters TEC used for scseq analysis, since the extreme low number of the cells wouldn't provide enough power. Also, I appreciate the explanation for the specific experimental process used in validation of genes from sequencing technologies, ie. Mixing PBMC and thymocytes. I think that the authors addressed all my concerns and clarified the issues that caused my confusion.

Thank you very much for the thoughtful comments. We convince that feedbacks from the referee greatly contributed to the improvement of the manuscripts.